# FSNet: Feasibility-Seeking Neural Network for Constrained Optimization with Guarantees

**Hoang T. Nguyen**
Massachusetts Institute of Technology
Cambridge, MA, USA
hoangh@mit.edu

**Priya L. Donti**
Massachusetts Institute of Technology
Cambridge, MA, USA
donti@mit.edu

## Abstract

Efficiently solving constrained optimization problems is crucial for numerous real-world applications, yet traditional solvers are often computationally prohibitive for real-time use. Machine learning-based approaches have emerged as a promising alternative to provide approximate solutions at faster speeds, but they struggle to strictly enforce constraints, leading to infeasible solutions in practice. To address this, we propose the Feasibility-Seeking Neural Network (FSNet), which integrates a feasibility-seeking step directly into its solution procedure to ensure constraint satisfaction. This feasibility-seeking step solves an unconstrained optimization problem that minimizes constraint violations in a differentiable manner, enabling end-to-end training and providing guarantees on feasibility and convergence. Our experiments across a range of different optimization problems, including both smooth/nonsmooth and convex/nonconvex problems, demonstrate that FSNet can provide feasible solutions with solution quality comparable to (or in some cases better than) traditional solvers, at significantly faster speeds.[1]

## 1   Introduction

Machine learning (ML) has emerged as a promising technique to provide fast surrogate solvers for parametric optimization problems [1, 2]. Specifically, given a set of optimization problems that share a fixed structure but whose parameters vary between instances, ML can be used to learn approximate, fast-to-evaluate mappings from parameters to optimization solutions. With this approach, many expensive real-world optimization problems can now be solved using ML in a fraction of a second, with promising examples in electric power systems [3, 4], building energy management systems [5], communication networks [6], robotic planning and control [7, 8], and supply chain management [9]. Unfortunately, although fast at producing solutions, ML approaches often struggle to satisfy problem constraints, resulting in solutions that are infeasible and potentially unusable, particularly in safety-critical applications. While several constraint-enforcement approaches have been proposed to mitigate this problem, including penalty methods [10], projection-based methods [11, 12], and iterative methods [13, 14], these methods often suffer from poor solution quality, expensive computational costs, and/or lack of guarantees.

To address these challenges, we propose the **F**easibility-**S**eeking Neural **N**etwork (**FSNet**), a general framework for solving constrained optimization problems using neural networks (NNs). Our key idea is to incorporate a differentiable feasibility-seeking step into the NN's training and inference processes to minimize constraint violations while still enabling end-to-end training. In particular, this feasibility-seeking step uses the NN's prediction as the starting point for a constraint violation minimization problem, which is solved using an iterative optimization solver to ultimately produce

---

[1]The code is available at https://github.com/MOSSLab-MIT/FSNet

39th Conference on Neural Information Processing Systems (NeurIPS 2025).

a feasible solution. This solution is evaluated via an unsupervised loss function that captures the objective function of the target optimization problem and the distance between the pre- and post-feasibility-seeking points. The full procedure is trained end-to-end, including backpropagation through the unrolled iterations of the optimization solver used in the feasibility-seeking step.

Our main contributions are summarized as follows:

- **General framework for constraint enforcement in NNs.** We introduce a principled and general framework, FSNet, that enables NNs to predict solutions satisfying both equality and inequality constraints in continuous optimization problems. The framework accommodates both convex and nonconvex constraints, making it applicable to many problem domains.

- **Theoretical guarantees on feasibility and convergence.** Under boundedness, $L$-smoothness, and the Polyak–Łojasiewicz (PL) condition, FSNet guarantees exponentially fast convergence to a feasible point and provably reaches a stationary point in the parameter space when being trained using stochastic gradient descent, even when backpropagation is truncated. We also specify conditions under which FSNet's predictions provably coincide with a true minimizer of the original constrained optimization problem.

- **Strong empirical performance across various classes of problems.** Across convex and nonconvex problems, FSNet produces feasible solutions and achieves small optimality gaps (less than 0.2% in convex cases), while providing solutions orders-of-magnitude faster than traditional solvers. In nonconvex cases, we show that FSNet can even find better local solutions than traditional solvers, at significantly greater speed.

## 2 Related Work

**ML for optimization.** There has been a large body of literature leveraging ML to provide fast solutions to optimization problems, split into two major classes of approaches (see also reviews [4, 15, 16]). The first class of approaches entails using ML to learn end-to-end surrogate models, in which problem parameters that vary between optimization problem instances are treated as inputs to the ML model [1, 2]. This includes supervised learning methods that require a training dataset of solved instances generated via traditional solvers [10, 17, 18], as well as self-supervised/unsupervised approaches that directly embed the optimization objective and/or constraint functions into the ML loss function [3, 13, 19, 20]. In this work, we adopt the self-supervised paradigm, which has been favored in recent work as it avoids the large amount of time and resources required to create supervised training datasets [21]. The second class of ML-for-optimization approaches involves using ML to improve the performance of other optimization solvers. For instance, ML has been used to approximate algorithm steps for convex optimization to achieve guaranteed fast convergence [22], provide warm start points to accelerate fixed-point optimization algorithms [23], or approximate or remove complex or redundant constraints to reduce computational complexity [24, 25]. The choice between whether to use an end-to-end vs. ML-for-other-solvers approach has historically been philosophical, with a lack of rigorous comparison. While our work is primarily situated within the end-to-end surrogates literature, we also draw inspiration from the ML-for-other-solvers approach, notably by incorporating an optimization warm-start component within our framework.

**Feasibility enforcement in ML-for-optimization.** While end-to-end ML approaches have been successful in providing fast solutions, these approaches have struggled to satisfy problem constraints. Several constraint-enforcement approaches have been proposed to mitigate this, including penalty methods, projection-based methods, and iterative methods. Penalty methods incentivize constraint enforcement by penalizing constraint violations within the ML loss function [10, 26]; however, these methods do not have feasibility guarantees and can generate highly infeasible solutions in practice [3, 13, 26]. Projection-based methods *do* provide feasibility guarantees, by projecting the output of the ML model onto the feasible set [11, 12]—in the context of NNs, this can be done via differentiable orthogonal projection methods that admit end-to-end training [11, 20, 27]. In special cases where the projection admits a computationally efficient special structure (e.g., solvable in closed form), projection methods can work extremely well, providing fast, feasible, high-quality solutions [19, 28, 29]. However, in general cases, solving orthogonal projection problems can be expensive, making training and inference slow, particularly in high-dimensional problems. Iterative methods aim to be more computationally efficient by instead deploying iterative algorithms to feasibility-correct the predictions of an ML algorithm. Notable examples include DC3 [13], which uses a variable

elimination-based approach to satisfy equality constraints and a gradient-based iterative approach to satisfy inequality constraints, and ENFORCE [14], which handles equality constraints by unrolling a sequential quadratic programming procedure. Our work presents an iterative method that handles both equality and inequality constraints and, unlike previous iterative methods, provides guarantees on constraint enforcement, which are particularly important in (e.g.) safety-critical settings.

**Differentiable optimization in deep learning.** Optimization problems can be directly integrated into neural networks as differentiable layers to embed relevant inductive biases or constraint satisfaction directly into the learning process. To enable end-to-end training, gradients must be computed through these optimization layers. *Implicit differentiation* approaches compute the gradient implicitly using optimality conditions or fixed-point equations, with approaches presented in the literature for, e.g., quadratic programs [20, 30], disciplined convex problems [27], cone problems [31, 32], submodular minimization [33], and other problem classes [34]. *Unrolled differentiation*, by contrast, explicitly unrolls the optimization iterations and leverages automatic differentiation tools to track gradients through each step [35–38]. While implicit differentiation can be more computationally efficient and numerically stable, it relies on obtaining an optimal solution to the optimization problem (which may be unavailable if the optimization did not converge) and is only applicable in situations where the ML-generated inputs show up in the fixed-point equations (which is not true for, e.g., warm start points). For the latter reason, FSNet employs unrolled differentiation (with truncated gradients) to differentiate through the feasibility-seeking step during training. For a review of methods integrating optimization within neural networks, as well as differentiation techniques, we refer readers to [39, 40].

## 3 FSNet: Feasibility-Seeking Neural Network

We now present FSNet, a novel NN-based framework for providing fast, feasible, high-quality solutions to constrained optimization problems. Specifically, we consider the task of solving parametric optimization problems of the form:

$$\min_{y \in \mathbb{R}^n} \quad f(y; x) \;\; \text{s.t.} \;\; g(y; x) \leq 0, \; h(y; x) = 0, \tag{1}$$

where $y \in \mathbb{R}^n$ are decision variables, $x \in \mathbb{R}^d$ are parameters, $f : \mathbb{R}^n \times \mathbb{R}^d \to \mathbb{R}$ is the objective function, and $g : \mathbb{R}^n \times \mathbb{R}^d \to \mathbb{R}^{n_{\text{ineq}}}$, $h : \mathbb{R}^n \times \mathbb{R}^d \to \mathbb{R}^{n_{\text{eq}}}$ are inequality and equality constraint functions, respectively. For illustrative examples of how $x$ and $y$ are defined in various practical problems, we refer the reader to the tutorial on amortized optimization [2]. Since the minimizer(s) of problem (1) change as the parameters $x$ vary, this implies an underlying mapping (or mappings) from the parameters $x$ to the minimizer(s). Thus, we can frame the problem of solving the parametric optimization problem as a learning problem in which we aim to use an ML model to approximate an underlying mapping. Specifically, let $\hat{y}_\theta : \mathbb{R}^d \to \mathbb{R}^n$ denote an NN-based function parameterized by $\theta \in \mathbb{R}^{n_\theta}$, with $\hat{y}_\theta(x)$ being its prediction for optimization parameters $x$. We aim to find $\theta$ by solving:

$$\min_{\theta \in \mathbb{R}^{n_\theta}} \quad \mathbb{E}_{x \sim \mathcal{D}}[f(\hat{y}_\theta(x); x)] \;\; \text{s.t.} \;\; g(\hat{y}_\theta(x); x) \leq 0, \; h(\hat{y}_\theta(x); x) = 0 \;\; \forall \, x \sim \mathcal{D}, \tag{2}$$

where $x$ follows an unknown distribution $\mathcal{D}$, and we assume access only to a finite dataset sampled from this distribution. Importantly, the goal is to choose $\theta$ to minimize the expected objective value while *ensuring that predictions satisfy all constraints*. However, due to sources of error such as optimization error, representation error, and generalization error, NNs alone often fail to produce feasible solutions. To address this, it is necessary to have an additional module that can strictly enforce the feasibility of the predicted minimizer for any $x \sim \mathcal{D}$.

To address this problem, we introduce the FSNet framework, which integrates a feasibility-seeking step into the NN training and inference pipelines. This framework is illustrated in Figure 1, with corresponding pseudocode in Algorithm 1. FSNet consists of three main parts: **prediction**, **feasibility seeking**, and **projection-inspired loss**. Specifically, the prediction step entails employing any standard NN $y_\theta : \mathbb{R}^d \to \mathbb{R}^n$ to map from input parameters $x$ to a candidate minimizer $y_\theta(x)$. As this candidate generally violates the constraints, we subsequently execute a feasibility-seeking procedure, which solves an infeasibility-minimization problem to transform $y_\theta(x)$ into a feasible solution $\hat{y}_\theta(x)$. We train this prediction + feasibility-seeking pipeline end-to-end to minimize a loss function that penalizes both the original optimization objective $f$ and the distance between the original prediction $y_\theta(x)$ and the post-feasibility-seeking outputs $\hat{y}_\theta(x)$.

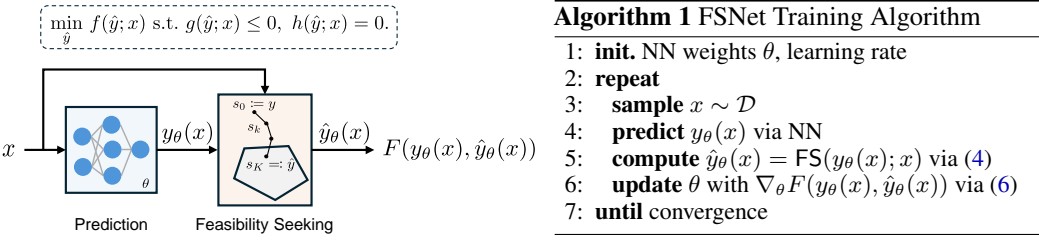

**Algorithm 1** FSNet Training Algorithm
1: **init.** NN weights $\theta$, learning rate
2: **repeat**
3:    **sample** $x \sim \mathcal{D}$
4:    **predict** $y_\theta(x)$ via NN
5:    **compute** $\hat{y}_\theta(x) = \mathsf{FS}(y_\theta(x); x)$ via (4)
6:    **update** $\theta$ with $\nabla_\theta F(y_\theta(x), \hat{y}_\theta(x))$ via (6)
7: **until** convergence

Figure 1: A schematic of FSNet, our framework for solving constrained optimization problems.

## 3.1 Feasibility-seeking step

We now discuss the idea and implementation of the feasibility-seeking step, which serves as the core mechanism in our framework for providing provable feasibility guarantees. Given a potentially infeasible NN prediction $y_\theta(x)$, one natural idea to ensure feasibility is to establish a process that starts at $y_\theta(x)$ and iteratively moves the point toward the feasible set. As this process progresses, the constraint violations are expected to gradually decrease and eventually vanish when a feasible point is reached. This idea can be formalized by leveraging $y_\theta(x)$ as a warm start point for the following minimization problem, and letting $\hat{y}_\theta(x)$ be the problem's obtained solution:

$$\min_{s \in \mathbb{R}^n} \ \phi(s; x) := \|h(s; x)\|_2^2 + \|g^+(s; x)\|_2^2, \tag{3}$$

where $\phi(s; x)$ quantifies the total violation of equality and inequality constraints for a given prediction $s$ and parameter $x$, and $g^+(s; x) = \max(g(s; x), 0)$ is applied element-wise. In practice, the violation-minimization problem (3) can be solved using one of many iterative optimization methods with local and/or global convergence guarantees [41]. For the purposes of our later theoretical analysis, we present the vanilla gradient descent method here, which minimizes $\phi(s; x)$ over iterations $k$ via:

$$s_0 = y_\theta(x), \quad s_{k+1} = s_k - \eta_\phi \nabla_s \phi(s_k; x), \tag{4}$$

where the iterations start from the prediction $y_\theta(x)$ and move in the direction of steepest descent with properly-chosen step size $\eta_\phi$. Let $\mathsf{FS} : \mathbb{R}^n \times \mathbb{R}^d \mapsto \mathbb{R}^n$ be a mapping that maps the initial point $y_\theta(x)$ and parameters $x$ to the final prediction $\hat{y}_\theta(x)$, in particular $\hat{y}_\theta(x) = \mathsf{FS}(y_\theta(x); x) := \lim_{k \to \infty} s_k$. In practice, we terminate our iterative procedure after a finite number of iterations until the function value $\phi(s_k; x)$ or its gradient drops below a small, pre-specified threshold.

To enable end-to-end training through the NN prediction and feasibility-seeking steps, we apply backpropagation through the unrolled iterates of the feasibility-seeking optimization process. (Since the prediction $y_\theta(x)$ does not appear within the optimality conditions of (3), implicit differentiation is not applicable for computing this gradient.)

Unlike projection-based methods that solve expensive constrained optimization problems, the feasibility-seeking step solves an unconstrained problem, which is typically easier and faster to solve. Unlike DC3 [13], which handles equality and inequality constraints separately, the feasibility-seeking step handles both simultaneously, which facilitates establishing feasibility guarantees (Section 4).

## 3.2 Loss function

We train the NN end-to-end through the feasibility-seeking procedure by minimizing an unsupervised empirical loss function over a dataset $\{x^{(1)}, \dots, x^{(S)}\}$, defined as follows:

$$\mathcal{L}(\theta) = \frac{1}{S} \sum_{i=1}^{S} F\left(y_\theta(x^{(i)}), \hat{y}_\theta(x^{(i)})\right), \tag{5}$$

where $\hat{y}_\theta(x) = \mathsf{FS}(y_\theta(x); x)$ is the output of the feasibility-seeking step, and we define the function $F(y_\theta(x), \hat{y}_\theta(x))$ as:

$$F(y_\theta(x), \hat{y}_\theta(x)) = f(\hat{y}_\theta(x); x) + \frac{\rho}{2} \|y_\theta(x) - \hat{y}_\theta(x)\|_2^2, \tag{6}$$

with penalty weight $\rho > 0$. The term $f(\hat{y}_\theta(x); x)$ promotes low objective value, while the term $\|y_\theta(x) - \hat{y}_\theta(x)\|_2^2$ encourages minimal corrections by the feasibility-seeking step, thereby incentivizing the NN to produce predictions that are closer to feasible. From a geometric perspective, the term $\|y_\theta(x) - \hat{y}_\theta(x)\|_2^2$ can be interpreted as a soft approximation of a projection, loosely corresponding to finding a feasible point $\hat{y}_\theta(x)$ that remains close to the original prediction $y_\theta(x)$. This term also plays a critical role in the convergence analysis of our framework, which we describe in the next section.

## 4  Convergence Analysis

In solving problem (1), FSNet is expected to predict a solution that minimizes the objective function while satisfying all constraints. Toward showing this, we prove that the feasibility-seeking procedure finds a feasible point (if one exists) at an exponentially fast rate (Section 4.1) and that the overall FSNet training procedure converges in the sense of the expected gradient norm vanishing asymptotically (Section 4.2). Finally, we characterize a condition under which FSNet's prediction $\hat{y}_\theta(x)$ coincides with an optimal solution of the target problem (Section 4.3). All proofs are given in Appendix D.

### 4.1  Convergence of feasibility-seeking step

As the feasibility-seeking step essentially solves an unconstrained optimization problem via gradient descent, the convergence analysis of the feasibility-seeking step reduces to an analysis of the performance of gradient descent in unconstrained optimization. We state the relevant theorem and associated standard assumption below, and refer the reader to [42, 43] for full proofs.

**Assumption 1.** *For any $x$, function $\phi(\cdot\,; x)$ is $L_\phi$-smooth, admits a minimizer $s^\star \in \mathbb{R}^n$, and satisfies the PL condition with constant $\mu_\phi$, i.e., $\|\nabla\phi(s; x)\|_2^2 \geq 2\mu_\phi\big(\phi(s; x) - \phi(s^\star; x)\big)$.*

**Theorem 1** ([42])**.** *Let $\gamma = 1 - \mu_\phi/L_\phi$. Under Assumption 1, the sequence $\{s_k\}$ generated by the gradient descent (4) with step size $\eta_\phi = 1/L_\phi$ converges to $s^\star$ with rate*

$$\|s_k - s^\star\| \leq \frac{8\eta_\phi L_\phi^2}{\mu_\phi^2}\gamma^{k-1}(\phi(s_0; x) - \phi(s^\star; x)), \tag{7}$$

$$\phi(s_k; x) - \phi(s^\star; x) \leq \gamma^k(\phi(s_0; x) - \phi(s^\star; x)). \tag{8}$$

From Theorem 1, we see that if the target optimization problem (1) admits at least one feasible point for a given $x$, i.e., $\phi(s^\star; x) = 0$, then the feasibility-seeking step drives the NN's prediction $y_\theta(x)$ to a feasible point at an exponentially fast rate. Indeed, gradient descent needs at most $\mathcal{O}(\log(1/\epsilon_\phi))$ iterations to reduce the violation value $\phi(s_k; x)$ below a small threshold $\epsilon_\phi > 0$.

### 4.2  Convergence of NN training

This section analyzes the convergence of FSNet training using stochastic gradient descent (SGD). Before delving into the analysis, we emphasize an important practical aspect that can impact the convergence of training: the feasibility-seeking problem (3) is typically solved by unrolling a finite number of gradient descent steps in (4) until a small pre-specified convergence tolerance is met. To formalize this, let $\mathsf{FS}^K(y_\theta(x); x)$ denote the feasibility-seeking mapping obtained by unrolling gradient descent in (4) for $K$ iterations. We then define the resulting approximate solution as $\hat{y}_\theta^K(x) \coloneqq \mathsf{FS}^K(y_\theta(x); x)$. Under this finite unrolling scheme, we now introduce a modified loss function to reflect the approximate nature of the feasibility-seeking step:

$$L(\theta) = \frac{1}{S}\sum_{i=1}^{S} F\big(y_\theta(x^{(i)}), \hat{y}_\theta^K(x^{(i)})\big). \tag{9}$$

The NN training is performed using SGD with an unbiased gradient estimator $\tilde{\nabla}L(\theta_t)$:

$$\theta_{t+1} = \theta_t - \eta\tilde{\nabla}L(\theta_t) \quad \text{with} \quad \mathbb{E}_t[\tilde{\nabla}L(\theta_t)] = \nabla L(\theta_t), \tag{10}$$

where $\mathbb{E}_t[\cdot]$ is the conditional expectation given the randomness up to step $t$.

We now analyze the convergence of NN training under this finite unrolling scheme. Establishing SGD convergence for the modified loss $L$ follows standard arguments [44]. However, to understand

how finite unrolling of the feasibility-seeking step affects optimization of the exact loss $\mathcal{L}$, we must also bound the discrepancy $\|\nabla L(\theta) - \nabla \mathcal{L}(\theta)\|_2$. Thus, we need to make boundedness and $L$-smoothness assumptions to control this discrepancy; otherwise, it can be arbitrarily large. Below, we use $J_w \psi(w) := \partial \psi(w)/\partial w$ to denote the Jacobian of vector-valued function $\psi$ with respect to $w$.

**Assumption 2** (Boundedness). *There exist constants $B_N, B_{FS}, B_F > 0$ such that for all $\theta, x$:*

$$\|J_\theta y_\theta(x)\|_2 \le B_N, \quad \|J_y \mathsf{FS}^K(y_\theta(x); x)\|_2 \le B_{FS}, \quad \|\nabla_{\hat{y}} F(y_\theta(x), \hat{y}_\theta(x))\|_2 \le B_F.$$

**Assumption 3** (Smoothness). *The function $F(y, \hat{y})$ is $L_F$-smooth with respect to $\hat{y}$, and the function $L(\theta)$ is $L_N$-smooth and bounded below by $L^\star$. The mappings $\mathsf{FS}$ and $\mathsf{FS}^K$ satisfy for $L_{FS} > 0$ that*

$$\|J_y \mathsf{FS}(y_\theta(x); x) - J_y \mathsf{FS}^K(y_\theta(x); x)\|_2 \le L_{FS}\|\hat{y}_\theta(x) - \hat{y}_\theta^K(x)\|_2, \quad \forall x \in \mathbb{R}^d.$$

**Theorem 2.** *Suppose that $\mathbb{E}_t \left[ \|\tilde{\nabla} L(\theta_t)\|_2^2 \right] \le G$ for all $t$ and that the feasibility-seeking step is unrolled for $K$ steps in every forward pass. Let $\gamma$ be as defined in Theorem 1. Under Assumption 1, 2, and 3, after running the SGD for $T$ iterations with step size $\eta = \eta_0/\sqrt{T}$ for some $\eta_0 > 0$, there is at least one iteration $t \in \{0, \ldots, T-1\}$ satisfying*

$$\mathbb{E} \left[ \|\nabla L(\theta_t)\|_2^2 \right] \le \mathcal{O} \left( T^{-1/2} \right), \tag{11}$$

$$\mathbb{E} [\|\nabla \mathcal{L}(\theta_t)\|_2] \le \mathcal{O} \left( T^{-1/4} + \gamma^K \right). \tag{12}$$

This theorem shows that, when the finite unrolling is done in practice, SGD converges in expectation to a stationary point of the NN parameters. Moreover, the finite unrolling introduces a bias of order $\mathcal{O}(\gamma^K)$ to the decay rate of $\nabla \mathcal{L}$, but the bias decays exponentially in $K$ due to $\gamma < 1$. In other words, for a moderate number of unrolled steps $K$, the bias is negligible, and the gradient norms under finite and infinite unrolling diminish at virtually the same rate. Consequently, finite unrolling is not only computationally efficient but also sufficient to preserve convergence guarantees.

## 4.3 Convergence to minimizers

In the previous section, we analyzed parameter-level convergence of the FSNet training process. Here, we turn to output-level convergence: first, we derive conditions under which the network's prediction is a stationary point of the function $F$, and then we identify when that prediction exactly matches the minimizer of the target optimization problem (1).

By the chain rule, the gradient of the loss (5) is the mean of $(J_\theta y_\theta(x^{(i)}))^\top \nabla_y F \left( y_\theta(x^{(i)}), \hat{y}_\theta(x^{(i)}) \right)$ over all samples. This implies that SGD can stall for two reasons: either the gradient $\nabla_y F \left( y_\theta(x^{(i)}), \hat{y}_\theta(x^{(i)}) \right)$ vanishes, or it remains nonzero but lies in the nullspace of $J_\theta y_\theta(x^{(i)})$. The former means that the prediction of the NN is a stationary point of $F \left( y_\theta(x^{(i)}), \hat{y}_\theta(x^{(i)}) \right)$, which is desirable, since only a stationary point has a chance to be a minimizer of the target optimization problem (1). Therefore, we want to rule out the latter case. One potential approach is employing a sufficiently expressive NN so that Jacobian matrices are full-rank and stationary points of $F$ can be reached for all training inputs $x$. We formalize this observation in the following lemma.

**Lemma 1.** *Suppose SGD converges to $\theta^\star$, where $J(\theta^\star) := \left( (J_\theta y_{\theta^\star}(x^{(1)}))^\top \; \ldots \; (J_\theta y_{\theta^\star}(x^{(S)}))^\top \right)$ has full column rank. Then, the prediction of the NN $y_\theta(x^{(i)})$ is the stationary point of $F(y_\theta(x^{(i)}), \hat{y}_\theta(x^{(i)}))$ with $\hat{y}_\theta(x^{(i)}) = \mathsf{FS}(y_\theta(x^{(i)}); x^{(i)})$ for all $i \in \{1, \ldots, S\}$.*

This lemma means that after being successfully trained, the NN in the FSNet framework can predict a stationary point $y_\theta(x)$ of $F(y_\theta(x), \hat{y}_\theta(x))$. If $F$ is convex, then the stationary point is a global minimizer of $F$. However, if $F$ is nonconvex, the stationary point could be a minimizer, maximizer, or saddle point. In such a case, additional assumptions are necessary for further analysis. Thus, we now make a (fairly strong) assumption that the NN's prediction is a global minimizer of $F$. However, a minimizer of $F$ may not necessarily be a minimizer of the target problem (1), because $F$ differs from the original objective function $f$ in including a penalty term. Inspired by the penalty method in constrained optimization [41], we can increase the penalty weight $\rho$ such that the difference $\|y_\theta(x) - \hat{y}_\theta(x)\|_2$ vanishes, which drives the minimizers of $F$ and problem (1) to coincide.

**Theorem 3.** *Given specific parameters $x$, suppose that the NN predicts $y_\tau$ which is a global minimizer of $F(y, \hat{y}; \rho_\tau) = f(\hat{y}; x) + \frac{\rho_\tau}{2}\|y - \hat{y}\|_2^2$ with $\hat{y} = \mathsf{FS}(y; x)$; that the mapping $\mathsf{FS}$ is continuous; and that $0 < \rho_\tau < \rho_{\tau+1}, \forall \tau$ and $\rho_\tau \to \infty$. Then, every limit point $(y^\star, \hat{y}^\star)$ of the sequence $\{(y_\tau, \hat{y}_\tau)\}$ admits $y^\star = \hat{y}^\star$, and $\hat{y}^\star$ is a global minimizer of the optimization problem (1).*

We acknowledge that the assumption in Theorem 3 is rather stringent, but it nonetheless provides some useful insights for practical implementation. In particular, the penalty term $\frac{\rho}{2}\|y_\theta(x) - \hat{y}_\theta(x)\|_2^2$ in (6) needs to be increasing to yield high-quality solutions. From our experiments, we find that choosing a sufficiently large value of $\rho$ is sufficient , and the range of effective $\rho$ is wide, making it easier to select an appropriate value.

## 5 Practical Efficiency Improvements

While the above FSNet framework offers theoretical guarantees, we also provide practical strategies to improve training efficiency and stability. We discuss the use of truncated gradient tracking to reduce computational overhead, and an enhancement to the loss function to improve training stability.

**Truncated unrolled differentiation.** Inspired by [37], we implement truncated unrolled differentiation to improve the computational efficiency of the FSNet framework. Specifically, when computing gradient updates, we backpropagate through only the first $K'$ out of the total $K$ iterations of the feasibility-seeking procedure, rather than all $K$ iterations. In other words, only the computation of the first $K'$ iterations is included in the computational graph for backpropagation, while the remaining iterations act as pass-through layers without gradient tracking. This approach significantly reduces computational graph size, memory, and the overhead of the backward pass [37]. We empirically observe negligible loss in solution quality even for modest $K'$, and in Appendix D.2, we prove that, under local strong convexity of $\phi$, the Jacobian error and the truncation-induced bias in the SGD updates decay exponentially with $K'$.

**Stabilizing early-stage training.** With randomly initialized parameters, the NN may produce large constraint violations early in training, forcing the feasibility-seeking step to use many iterations and slowing down overall training. To mitigate this, we introduce an additional penalty for constraint violation when it exceeds a threshold:

$$F(y_\theta(x), \hat{y}_\theta(x)) = \begin{cases} f(\hat{y}_\theta(x); x) + \frac{\rho}{2}\|y_\theta(x) - \hat{y}_\theta(x)\|_2^2 + \rho_\phi \phi(y_\theta(x); x) & \text{if } \phi(y_\theta(x); x) \geq Q, \\ f(\hat{y}_\theta(x); x) + \frac{\rho}{2}\|y_\theta(x) - \hat{y}_\theta(x)\|_2^2 & \text{otherwise,} \end{cases}$$
(13)

where the threshold $Q$ can be big, e.g., $Q = 10^3$. When constraint violations are too large ($\phi(y; x) \geq Q$), the extra term adds an additional penalty to the loss that pushes the NN to make more feasible predictions. Otherwise, the extra term is removed, and the loss reverts exactly to the original form.

## 6 Experiments

**Problem Classes.** We evaluate our method on three families of synthetic problems: *smooth convex*, *smooth nonconvex*, and *nonsmooth nonconvex*. The *smooth convex* problems include quadratic programs (QPs), quadratically-constrained quadratic programs (QCQPs), and second-order cone programs (SOCPs). *Smooth nonconvex* problems are constructed by adding elementwise sine/cosine terms into the objective and constraint functions of the smooth convex problems, while the *nonsmooth nonconvex* problems further add an $\ell_2$-norm regularization term to the objective function (Appendix A). Each problem has 100 decision variables subject to 50 equality and 50 inequality constraints. We generate a dataset of 10000 samples for each problem following the procedure in [13, 45], and split it into 7000 training, 1000 validation, and 2000 test samples for all NN-based methods. For the smooth convex settings, we additionally consider larger-scale problems with 500 decision variables, 200 equality constraints, and 200 inequality constraints. We use 17000 training, 1000 validation, and 2000 test samples for this case. Finally, we consider the problem of AC optimal power flow (ACOPF), a nonconvex, NP-hard problem in electric power systems that aims to schedule power generation at least cost while satisfying grid and device constraints; for this problem, we use the formulation and dataset from [46].

**Baselines.** We compare our FSNet against:

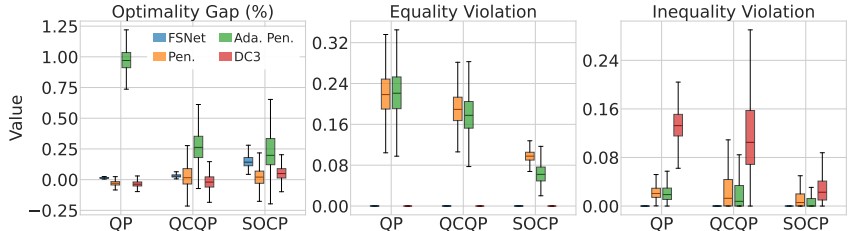

Figure 2: Test results on 2000 instances of convex problems with 100 decision variables. FSNet consistently obtains near-zero constraint violations and small optimality gaps across the problem classes, whereas the baseline methods incur significant constraint violations.

- **Solver:** A traditional solver. We use OSQP [47] and SCS [48] for convex problems, and use IPOPT [49] for nonconvex problems.
- **Reduced Solver:** For a fairer runtime comparison, we reduce the solvers' tolerances to achieve a level of constraint satisfaction comparable to FSNet.
- **Penalty:** A popular approach for addressing constraints in training NNs that adds a fixed-weight penalty for constraint violations within the loss function.
- **Adaptive Penalty:** An improved version of the fixed penalty method that adjusts the penalty weight based on current violations, as in [10].
- **DC3:** The Deep Constraint Completion and Correction framework from [13].

We also implement the **Projection** method in the convex QP setting, which orthogonally projects the NN's prediction onto the feasible set using the qpth solver [20]. While we attempted to implement Projection for other problem classes via cvxpylayers [27], this introduced significant computational overhead, making training prohibitively slow. All network, training, and solver-related hyperparameters are in Appendix C. We also provide experiments analyzing FSNet's performance with varying numbers of tracked iterations in truncated unrolled differentiation (Section 5), different values of the penalty weight $\rho$, and additional results for DC3 using our loss function (5) in Appendix B.

**Evaluation metrics.** We evaluate the methods using the following metrics: constraint violations, optimality gap, and computational time. We compute $\|h(\hat{y}_\theta(x); x)\|_1$ for equality violations and $\|g^+(\hat{y}_\theta(x); x)\|_1$ for inequality violations. Since some trusted solvers can return the global minimum for convex problems, we take their solution $y_{\text{solver}}$ to define the relative optimality gap by:

$$\text{optimality gap} = \frac{f(\hat{y}_\theta(x); x) - f(y_{\text{solver}}; x)}{|f(y_{\text{solver}}; x)|}.$$

For computational time, we compare two metrics: Batch and Sequential computational time. For Batch computational time, all problem instances were solved in a setting of maximal parallelization within a particular budget of compute hardware. For Sequential computational time, all instances were solved one-at-a-time in sequence. Traditional solvers were run on 1 Intel(R) Xeon(R) CPU E5-2670 v2 and all NN methods were run on 1 NVIDIA Quadro RTX 8000 GPU. All metrics are computed over 3 runs. Results on these metrics are summarized below, with full results in Appendix B.

**Feasibility-seeking method.** We employ L-BFGS with backtracking line search for feasibility seeking, as it nicely balances convergence rate and scalability to high-dimensional problems [41].

### 6.1 Smooth convex problems

For the convex settings with 100 decision variables, Figure 2 shows that FSNet outperforms the other baselines by obtaining near-zero constraint violations, whereas other methods incur significant constraint violations. For DC3, equality constraints are handled well, but inequality constraints are not satisfied across all cases; moreover, we observe that the performance of DC3 on the inequality constraints is sensitive to the step size of correction steps. In contrast, our feasibility-seeking step based on L-BFGS performs robustly across problem classes without needing to change the hyperparameters.

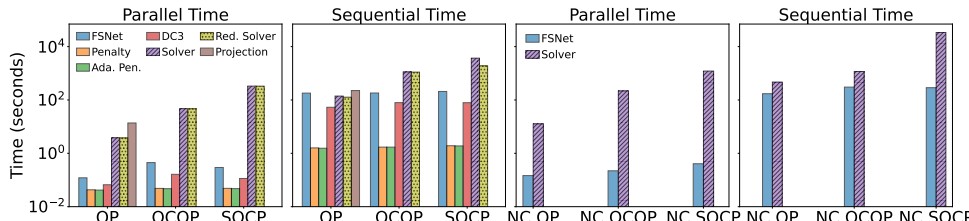

Figure 3: Computational time on 2000 instances of convex and nonconvex (NC) problems with 100 decision variables.

| Method | Equality Vio. Mean (Max) | Inequality Vio. Mean (Max) | Optimality Gap (%) Mean | Min (Max) | Runtime (s) Batch (sequential) |
|---|---|---|---|---|---|
| **Convex QP:** $n = 500, n_{\text{eq}} = 200, n_{\text{ineq}} = 200$ | | | | | |
| Solver | 1.9e-9$\pm$2.9e-10 (1.2e-6$\pm$9.8e-8) | 1.4e-6$\pm$9.8e-8 (9.3e-4$\pm$1.3e-4) | – | – | 112.962$\pm$0.378 (1394.024$\pm$94.868) |
| FSNet | 1.0e-4$\pm$2.5e-6 (4.4e-3$\pm$8.2e-4) | 4.5e-6$\pm$1.5e-7 (4.2e-4$\pm$9.0e-5) | 0.066$\pm$2.7e-4 | 0.041$\pm$6.6e-4 (0.119$\pm$4.9e-3) | 0.599$\pm$1.8e-3 (163.558$\pm$1.027) |
| **Convex QCQP:** $n = 500, n_{\text{eq}} = 200, n_{\text{ineq}} = 200$ | | | | | |
| Solver | 1.3e-5$\pm$4.3e-7 (3.1e-4$\pm$7.7e-5) | 5.0e-2$\pm$9.9e-4 (2.1e-1$\pm$7.3e-3) | – | – | 5508.419$\pm$44.220 (33362.147$\pm$107.952) |
| FSNet | 2.7e-4$\pm$1.4e-6 (3.7e-4$\pm$3.3e-7) | 9.9e-7$\pm$4.5e-8 (1.0e-5$\pm$4.9e-7) | 0.143$\pm$1.3e-3 | 0.087$\pm$2.5e-3 (0.328$\pm$0.034) | 37.291$\pm$0.237 (232.837$\pm$1.229) |
| **Convex SOCP:** $n = 500, n_{\text{eq}} = 200, n_{\text{ineq}} = 200$ | | | | | |
| Solver | 5.8e-13$\pm$5.2e-16 (2.1e-12$\pm$3.1e-13) | 2.2e-10$\pm$3.1e-13 (8.1e-8$\pm$2.7e-8) | – | – | 523.045$\pm$6.284 (490084.123$\pm$98143.538) |
| FSNet | 2.7e-4$\pm$1.4e-6 (3.7e-4$\pm$2.1e-6) | 0.0$\pm$0.0 (0.0$\pm$0.0) | 0.730$\pm$0.027 | 0.469$\pm$0.246 (1.124$\pm$0.048) | 14.313$\pm$0.073 (177.890$\pm$1.098) |

Table 1: Results of FSNet on 2000 instances of smooth convex problems with 500 decision variables.

FSNet also obtains small optimality gaps, all less than $0.2\%$ on average. The baseline methods also exhibit small optimality gaps, but their solutions violate the constraints significantly, which means these gaps are not meaningful. Specifically, we see that when constraint violations exist, NNs can easily find an infeasible point to achieve low objective values and optimality gaps— e.g., Penalty, Adaptive Penalty, and even DC3 attain negative optimality gaps for some instances (Figure 2).

Overall, FSNet achieves order-of-magnitude speedups over traditional solvers—e.g., 30-1000$\times$ faster in batch solving (Figure 3). For QP problems, FSNet outperforms OSQP when using batch solving but is slower under sequential solving; this difference arises because OSQP is specifically designed to exploit the problem structure of QPs, whereas FSNet does not yet incorporate such problem-specific optimizations. Despite constraint satisfaction, the projection method runs about 3.6$\times$ slower than the traditional solver, and about 113$\times$ slower than FSNet in the batch setting (Figure 3). In contrast, for QCQP and SOCP problems, FSNet consistently outperforms traditional solvers in both batch and sequential settings. Due to the incorporation of the feasibility-seeking step, FSNet has a slower runtime compared to other NN-based methods; however, this extra overhead is a necessary tradeoff to ensure the feasibility of the prediction. Conversely, faster methods fail to ensure feasible solutions, rendering their speed meaningless in applications where constraint satisfaction is non-negotiable.

For larger-scale convex problems (with 500 variables, 200 equality constraints, and 200 inequality constraints), the results are presented in Table 1. In the batch setting, FSNet achieves speedups of 188$\times$ (QP) and 147$\times$ (QCQP), which are notably higher than the speedups obtained in the smaller problems with 100 decision variables (namely, 31$\times$ and 104$\times$, respectively). For SOCP, FSNet achieves a smaller speedup of 36$\times$ compared to 1125$\times$ in the 100-variable case. In the sequential setting, FSNet achieved speedups of 8$\times$ (QP),143$\times$ (QCQP), and 2754$\times$ (SOCP), compared to 0.8$\times$, 6$\times$, and 18$\times$ in the corresponding 100-variable cases. In general, FSNet becomes increasingly advantageous in larger-scale settings. This can be explained by the fact that the feasibility-seeking step, which is an unconstrained feasibility problem and the dominant component of FSNet, is inherently more scalable than solving the original constrained optimization problems.

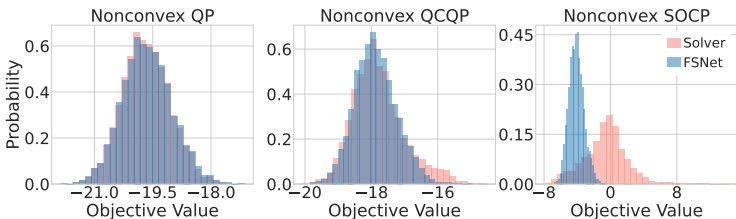

Figure 4: Distributions of objective values for solvers and FSNet in nonconvex problems. FSNet obtains sharper and lower-centered objective distributions for Nonconvex QCQP and Nonconvex SOCP, while matching the solver's optimality in Nonconvex QP with nearly identical distributions.

## 6.2 Smooth nonconvex problems

On smooth nonconvex problems, FSNet consistently achieves near-zero constraint violations across nonconvex problems and achieves 85-3000× batch speedups and 2.7-117× sequential speedups over traditional solvers (Figure 3). While the computational time of FSNet remains relatively stable across problem types, nonconvex solvers typically require more time than their convex counterparts, resulting in even greater speedup gains for FSNet in nonconvex settings. As in the convex cases, the Penalty and Adaptive Penalty show poor performance on these problems with respect to constraint satisfaction. Results for DC3 are not shown, as this method sometimes diverged in the presence of nonconvex constraints due to the divergence of Newton's method in the completion procedure.

Notably, FSNet actually outperforms the traditional solvers in terms of optimality for nonconvex problems. Specifically, for the Nonconvex QCQP and Nonconvex SOCP problem variants, the optimality gaps are negative (Table B.2), indicating that the solutions predicted by FSNet yield lower objective values than those obtained by the solvers. The difference is clearly illustrated in Figure 4, where the distribution of FSNet objective values is sharper than that of the solver in Nonconvex QCQP, and FSNet's distribution for Nonconvex SOCP is tightly centered at lower objective values. For the Nonconvex QP, the two distributions are almost perfectly aligned, suggesting that both methods achieve similar solution quality.

## 6.3 Nonsmooth nonconvex problems and AC optimal power flow

Results on nonsmooth nonconvex problems and ACOPF problems are in Appendix B. The findings are generally consistent with those for smooth nonconvex problems. In particular, FSNet achieves near-zero constraint violations across these problems. For nonsmooth nonconvex problems, FSNet achieves 152-3316× speedups in the batch setting and 7-80× speedups in the sequential setting compared to the traditional solver. We similarly observe that FSNet can find better local solutions with lower objective values than the traditional solver in the nonsmooth nonconvex setting. For the ACOPF problem, FSNet is faster than IPOPT, achieving 3× speedups on the IEEE 30-bus system and 11× on the IEEE 118-bus system in the sequential computation, with a less than 1% optimality gap.

# 7    Conclusion

We proposed a novel framework, FSNet, that integrates a feasibility-seeking step into NN training to solve general constrained optimization problems, with theoretical guarantees on feasibility and convergence. The feasibility-seeking step solves an unconstrained optimization over a function of constraint violations, initialized at the NN's prediction, to produce a feasible solution. Our experiments on smooth/nonsmooth and convex/nonconvex problems demonstrate that FSNet consistently outperforms popular baseline methods in achieving negligible constraint violations. Regarding optimality, FSNet matches the performance of traditional solvers on convex problems and often finds better local solutions in nonconvex cases, all while being significantly faster.

We observe through the experiments that FSNet can perform well for problems that violate the assumptions used in the convergence analysis (e.g., $L$-smoothness, PL condition). Thus, extending the convergence analysis in Section 4 to cover such cases is a potential direction for future work. While FSNet shows strong performance across diverse problem classes, scalability to extremely large-scale settings also remains an open direction. Another potential future direction is further scaling up the performance of the feasibility-seeking step using more efficient optimization algorithms.

## Acknowledgments

This work was supported by the U.S. National Science Foundation under award #2325956.

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

# A Problem formulations

## A.1 Convex problems

The following formulations share: $y \in \mathbb{R}^n, Q \in \mathbb{S}_{++}^n, p \in \mathbb{R}^n, A \in \mathbb{R}^{n_{eq} \times n}, x \in \mathbb{R}^{n_{eq}}, L, U \in \mathbb{R}^n$.

$$\textbf{QP:} \quad \min_{L \leq y \leq U} \frac{1}{2} y^\top Q y + p^\top y$$
$$\text{s.t. } Ay = x, \quad Gy \leq h,$$

where $G \in \mathbb{R}^{n_{ineq} \times n}, h \in \mathbb{R}^{n_{ineq}}$.

$$\textbf{QCQP:} \quad \min_{L \leq y \leq U} \frac{1}{2} y^\top Q y + p^\top y$$
$$\text{s.t. } Ay = x, \quad y^\top H_i y + g_i^\top y \leq h_i,$$

where $H_i \in \mathbb{S}_{++}^n, g_i \in \mathbb{R}^n, h \in \mathbb{R}$ for $i = 1, \ldots, n_{ineq}$.

$$\textbf{SOCP:} \quad \min_{L \leq y \leq U} \frac{1}{2} y^\top Q y + p^\top y$$
$$\text{s.t. } Ay = x, \quad \|G_i y + h_i\|_2 \leq c_i^\top y + d_i,$$

where $G_i \in \mathbb{R}^{m \times n}, h_i \in \mathbb{R}^m, c_i \in \mathbb{R}^n, d_i \in \mathbb{R}$ for $i = 1, \ldots, n_{ineq}$.

## A.2 Nonconvex problems

We modify the convex problems by adding element-wise sine and cosine functions.

$$\textbf{Nonconvex QP:} \quad \min_{L \leq y \leq U} \frac{1}{2} y^\top Q y + p^\top \sin(y)$$
$$\text{s.t. } Ay = x, \quad G \sin(y) \leq h \cos(x),$$
$$\textbf{Nonconvex QCQP:} \quad \min_{L \leq y \leq U} \frac{1}{2} y^\top Q y + p^\top \sin(y)$$
$$\text{s.t. } Ay = x, \quad y^\top H_i y + g_i^\top \cos(y) \leq h_i,$$
$$\textbf{Nonconvex SOCP:} \quad \min_{L \leq y \leq U} \frac{1}{2} y^\top Q y + p^\top \sin(y)$$
$$\text{s.t. } Ay = x, \quad \|G_i \cos(y) + h_i\|_2 \leq c_i^\top y + d_i.$$

## A.3 Nonsmooth nonconvex problems

Compared to the (smooth) convex problems, we further add an $\ell_2$-norm regularization term to the objective function.

$$\textbf{Nonsmooth nonconvex QP:} \quad \min_{L \leq y \leq U} \frac{1}{2} y^\top Q y + p^\top \sin(y) + \lambda \|y\|_2$$
$$\text{s.t. } Ay = x, \quad G \sin(y) \leq h \cos(x),$$
$$\textbf{Nonsmooth nonconvex QCQP:} \quad \min_{L \leq y \leq U} \frac{1}{2} y^\top Q y + p^\top \sin(y) + \lambda \|y\|_2$$
$$\text{s.t. } Ay = x, \quad y^\top H_i y + g_i^\top \cos(y) \leq h_i,$$
$$\textbf{Nonsmooth nonconvex SOCP:} \quad \min_{L \leq y \leq U} \frac{1}{2} y^\top Q y + p^\top \sin(y) + \lambda \|y\|_2$$
$$\text{s.t. } Ay = x, \quad \|G_i \cos(y) + h_i\|_2 \leq c_i^\top y + d_i.$$

## A.4 AC optimal power flow

For the AC optimal power flow problem, we use the formulation from [46], reproduced below:

$$\min_{p^{\mathrm{g}},q^{\mathrm{g}},p^{\mathrm{f}},q^{\mathrm{f}},p^{\mathrm{t}},q^{\mathrm{t}},v,\theta} \quad \sum_{i\in\mathcal{N}}\sum_{j\in\mathcal{G}_i} c_j p_j^{\mathrm{g}}$$

$$\text{s.t.} \quad p_i^{\mathrm{g}} - p_i^{\mathrm{d}} - g_i^{\mathrm{s}} v_i^2 = \sum_{e\in\mathcal{E}_i} p_e^{\mathrm{f}} + \sum_{e\in\mathcal{E}_i^R} p_e^{\mathrm{t}} \qquad \forall i\in\mathcal{N}$$

$$q_i^{\mathrm{g}} - q_i^{\mathrm{d}} + b_i^{\mathrm{s}} v_i^2 = \sum_{e\in\mathcal{E}_i} q_e^{\mathrm{f}} + \sum_{e\in\mathcal{E}_i^R} q_e^{\mathrm{t}} \qquad \forall i\in\mathcal{N}$$

$$p_e^{\mathrm{f}} = g_e^{\mathrm{tt}} v_i^2 + g_e^{\mathrm{ft}} v_i v_j \cos(\theta_i - \theta_j) + b_e^{\mathrm{ft}} v_i v_j \sin(\theta_i - \theta_j) \qquad \forall e = (i,j)\in\mathcal{E}$$

$$q_e^{\mathrm{f}} = -b_e^{\mathrm{tt}} v_i^2 - b_e^{\mathrm{ft}} v_i v_j \cos(\theta_i - \theta_j) + g_e^{\mathrm{ft}} v_i v_j \sin(\theta_i - \theta_j) \qquad \forall e = (i,j)\in\mathcal{E}$$

$$p_e^{\mathrm{t}} = g_e^{\mathrm{tt}} v_j^2 + g_e^{\mathrm{tf}} v_i v_j \cos(\theta_i - \theta_j) - b_e^{\mathrm{tf}} v_i v_j \sin(\theta_i - \theta_j) \qquad \forall e = (i,j)\in\mathcal{E}$$

$$q_e^{\mathrm{t}} = -b_e^{\mathrm{tt}} v_j^2 - b_e^{\mathrm{tf}} v_i v_j \cos(\theta_i - \theta_j) - g_e^{\mathrm{tf}} v_i v_j \sin(\theta_i - \theta_j) \qquad \forall e = (i,j)\in\mathcal{E}$$

$$(p_e^{\mathrm{f}})^2 + (q_e^{\mathrm{f}})^2 \le \overline{S}_e^2 \qquad \forall e\in\mathcal{E}$$

$$(p_e^{\mathrm{t}})^2 + (q_e^{\mathrm{t}})^2 \le \overline{S}_e^2 \qquad \forall e\in\mathcal{E}$$

$$\underline{\Delta\theta}_e \le \theta_i - \theta_j \le \overline{\Delta\theta}_e \qquad \forall e = (i,j)\in\mathcal{E}$$

$$\theta^{\mathrm{ref}} = 0$$

$$\underline{p}_i^{\mathrm{g}} \le p_i^{\mathrm{g}} \le \overline{p}_i^{\mathrm{g}} \qquad \forall i\in\mathcal{G}$$

$$\underline{q}_i^{\mathrm{g}} \le q_i^{\mathrm{g}} \le \overline{q}_i^{\mathrm{g}} \qquad \forall i\in\mathcal{G}$$

$$\underline{v}_i \le v_i \le \overline{v}_i \qquad \forall i\in\mathcal{N}$$

$$-\overline{S}_e \le p_e^{\mathrm{f}} \le \overline{S}_e \qquad \forall e\in\mathcal{E}$$

$$-\overline{S}_e \le q_e^{\mathrm{f}} \le \overline{S}_e \qquad \forall e\in\mathcal{E}$$

$$-\overline{S}_e \le p_e^{\mathrm{t}} \le \overline{S}_e \qquad \forall e\in\mathcal{E}$$

$$-\overline{S}_e \le q_e^{\mathrm{t}} \le \overline{S}_e \qquad \forall e\in\mathcal{E}$$

where $\mathcal{N}$, $\mathcal{E}$, and $\mathcal{G}$ denote the sets of buses, branches, and generators, respectively; $p^{\mathrm{g}}$ and $q^{\mathrm{g}}$ represent the active and reactive generation powers; $p^{\mathrm{f}}$ and $q^{\mathrm{f}}$ ($p^{\mathrm{t}}$ and $q^{\mathrm{t}}$) denote the active and reactive power flows in the forward (reverse) direction; and $v$ and $\theta$ denote the voltage magnitude and angle. The network parameters $g^{\mathrm{ff}}, g^{\mathrm{ft}}, g^{\mathrm{tf}}, g^{\mathrm{tt}}$ and $b^{\mathrm{ff}}, b^{\mathrm{ft}}, b^{\mathrm{tf}}, b^{\mathrm{tt}}$ are obtained from the admittance matrix of the system. In addition, $\overline{S}$ denotes the apparent power flow limit, while $\underline{\Delta\theta}$ and $\overline{\Delta\theta}$ specify the lower and upper bounds on the voltage angle difference. The parameters $\underline{p}^{\mathrm{g}}$ and $\overline{p}^{\mathrm{g}}$ ($\underline{q}^{\mathrm{g}}$ and $\overline{q}^{\mathrm{g}}$) indicate the lower and upper limits of active (reactive) generation, and $\underline{v}$ and $\overline{v}$ define the permissible range of voltage magnitudes.

# B More experiment results

## B.1 Smooth convex problems

Table B.1 presents detailed results for smooth convex problems. Figure B.1 plots FSNet's runtime on 2000 test instances as a function of batch size. Thanks to GPU parallelism, the runtime decreases monotonically as batch size increases, so it is recommended to use the largest batch size possible to maximize speedup. Figure B.2 reports CPU utilization for the parallel solver runs. Across all problems, CPU usage remains near 100% throughout execution, showing that we fully exploit available CPU resources. This indicates that our runtime comparisons between NN-based methods and the solver are fair.

| Method | Equality Vio. Mean (Max) | Inequality Vio. Mean (Max) | Optimality Gap (%) Mean | Optimality Gap (%) Min (Max) | Runtime (s) Batch (Sequential) |
|---|---|---|---|---|---|
| **Convex QP:** $n = 100, n_{\text{eq}} = 100, n_{\text{ineq}} = 100$ | | | | | |
| Solver | 1.3e-13±1.7e-16 (1.9e-13±5.5e-15) | 2.4e-14±2.44e-16 (5.4e-14±4.19e-16) | – | – | 3.802±0.045 (140.328±4.585) |
| Reduced Solver | 1.2e-6±7.4e-8 (8.5e-5±1.0e-5) | 2.1e-4±1.1e-5 (7.1e-5±1.0e-5) | 2.5e-4±1.3e-5 | -1.7e-7±2.3e-7 (5.3e-3±3.9e-4) | 3.777±0.103 (127.105±5.372) |
| Projection (qpth) | 9.7e-14±3.5e-16 (1.4e-13±3.4e-15) | 1.2e-14±4.6e-16 (4.1e-13±2.2e-15) | 3.7e-3±1.7e-3 | 2.2e-4±1e-4 (0.02±5.3e-3) | 13.71±1.07 (225.063±43.585) |
| FSNet | 1.3e-4±1.1e-5 (7.7e-4±1.9e-4) | 1.6e-6±4.4e-7 (7.2e-5±3.2e-5) | 0.015±0.002 | 5.7e-3±9.7e-4 (0.038±7.7e-4) | 0.121±0.038 (180.389±22.505) |
| Penalty | 0.219±5.8e-3 (5.4e-5±2.1e-5) | 0.023±3.3e-3 (0.227±0.039) | -0.029±0.013 | -0.074±0.013 (0.028±9.8e-3) | 0.043±2.9e-3 (1.594±0.051) |
| Adaptive Penalty | 0.221±7.6e-3 (0.371±2.9e-3) | 0.022±1.6e-3 (0.211±0.019) | 0.974±0.059 | 0.782±0.064 (1.213±0.074) | 0.042±1.7e-3 (1.549±0.011) |
| DC3 | 4.9e-13±7.8e-16 (5.8e-13±6.8e-15) | 0.133±0.011 (0.259±0.021) | -0.035±0.016 | -0.087±0.019 (0.029±0.021) | 0.058±3.5e-3 (53.151±0.558) |
| **Convex QCQP:** $n = 100, n_{\text{eq}} = 50, n_{\text{ineq}} = 50$ | | | | | |
| Solver | 8.4e-7±1.9e-8 (2.4e-5±5.8e-6) | 1.5e-3±2.0e-5 (7.7e-3± 0.00) | – | – | 46.619±0.577 (1118.149±3.535) |
| FSNet | 1.2e-4±1.2e-5 (1.5e-3±5.7e-4) | 1.0e-6±2.2e-7 (6.4e-5±2.4e-5) | 0.035±3.2e-3 | 7.8e-3±1.1e-3 (0.359±0.069) | 0.448±0.016 (182.156±12.827) |
| Penalty | 0.191±5.9e-3 (0.355±0.014) | 0.031±0.01 (0.521±0.019) | 0.038±0.062 | -0.237±0.022 (0.397±0.2) | 0.048±3.9e-3 (1.707±6.8e-3) |
| Adaptive Penalty | 0.179±7.6±-3 (0.333±7.6e-3) | 0.028±2.1e-3 (0.497±0.075) | 0.274±0.042 | -0.119±0.067 (0.797±0.079) | 0.047±2.8e-3 (1.703±0.02) |
| DC3 | 3.0e-13±4.9e-16 (3.8e-13±6.4e-15) | 0.120±0.032 (0.602±0.072) | -0.018±0.037 | -0.198±0.041 (0.259±0.039) | 0.163±2.3e-3 (79.234±6.847) |
| **Convex SOCP:** $n = 100, n_{\text{eq}} = 50, n_{\text{ineq}} = 50$ | | | | | |
| Solver | 3.1e-14±4.2e-17 (2.2e-13±0.00) | 1.3e-11 ±1.2e-11 (1.0e-8±8.8e-9) | – | – | 332.109±2.186 (3702.308±9.56) |
| Reduced Solver | 5.9e-7±1.3e-8 ( 4.8e-6±3.3e-7) | 1.8e-4 ±4.1e-6 (2.0e-3±8.7e-5) | 4.8e-5±2.1e-6 | -1.2e-3±4.5e-5 (1.4e-3±1.1e-4) | 330.143±1.687 (1895.989±4.592) |
| FSNet | 1.3e-4±1.2e-5 (5.5e-4±9.9e-5) | 1.7e-8±1.3e-8 (4.9e-6±1.5e-6) | 0.159±5.0e-3 | 0.046±3.6e-3 (1.392±0.376) | 0.295±8.0e-3 (208.399±9.935) |
| Penalty | 0.098±5.2e-3 (0.139±3.6e-3) | 0.015±3.2e-3 (0.309±0.053) | 0.024±0.018 | -0.235±0.107 (0.338±0.108) | 0.049±3.4e-3 (1.919±0.236) |
| Adaptive Penalty | 0.064±2.5e-3 (0.136±7.2e-3) | 0.011±1.3e-3 (0.305±0.147) | 0.256±0.126 | -0.171±0.146 (0.808±0.359) | 0.047±1.4e-3 (1.891±2.239) |
| DC3 | 7.8e-14±6.9e-17 (1.1e-13±9.8e-16) | 0.029±6.0e-3 (0.325±0.049) | 0.053±0.011 | -0.103±9.5e-3 (0.304±0.064) | 0.114±5.4e-3 (78.83±5.254) |

Table B.1: Test results of on 2000 instances of smooth convex problems across 3 random seeds.

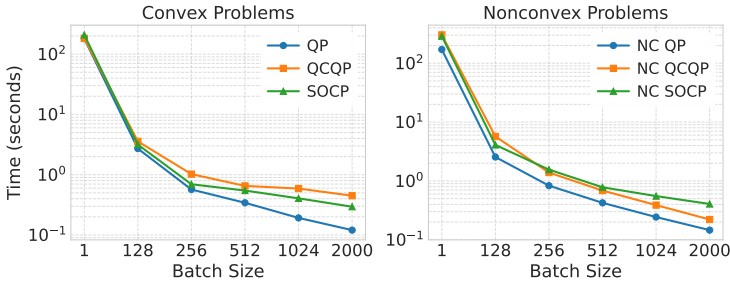

Figure B.1: Computational times of FSNet on 2000 instances of smooth convex and nonconvex (NC) problems with varying batch sizes.

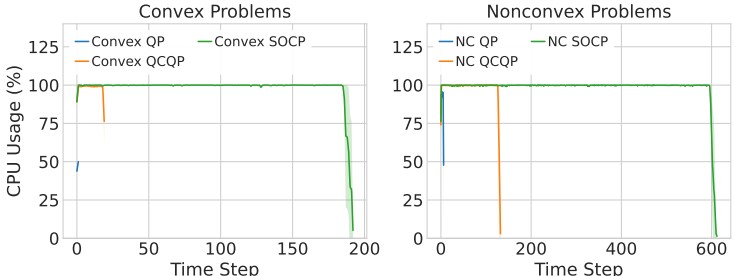

Figure B.2: CPU utilization over time for parallel solver runs on smooth convex and nonconvex problems.

## B.2   Smooth nonconvex problems

Figure B.3 and Table B.2 present the detailed results for nonconvex problems. Figure B.1 and B.2 also show the computational time of FSNet with different batch sizes and CPU utilization on nonconvex cases, respectively. The patterns are the same as those observed in the smooth convex problems.

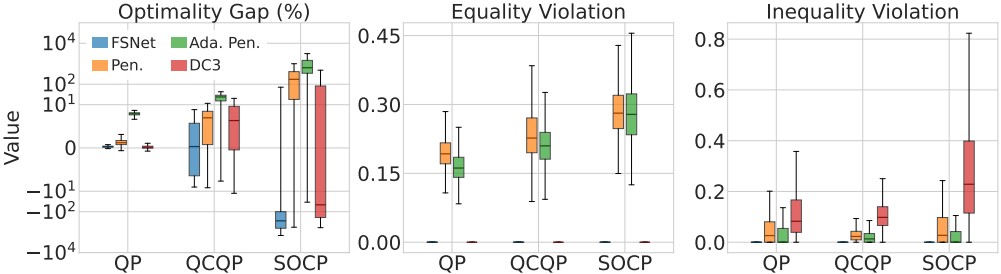

Figure B.3: Test results on 2000 instances of smooth nonconvex problems. FSNet consistently obtains near-zero constraint violations and small optimality gaps across the problem classes, whereas the baseline methods incur significant constraint violations.

| Method | Equality Vio. Mean (Max) | Inequality Vio. Mean (Max) | Optimality Gap (%) Mean | Optimality Gap (%) Min (Max) | Runtime (s) Batch (sequential) |
|---|---|---|---|---|---|
| | | | **Nonconvex QP:** $n = 100, n_{eq} = 100, n_{ineq} = 100$ | | |
| Solver | 5.5e-14±1.1e-16 (8.3e-14±5.1e-15) | 1.8e-9±7.7e-11 (1.6e-8±8.6e-10) | – | – | 12.699±0.692 (470.846±0.991) |
| FSNet | 9.3e-5±1.9e-6 (1.4e-3±2e-4) | 7.2e-7±8.7e-8 (4.4e-5±1.1e-5) | 0.146±0.012 | -0.341±0.24 (8.885±0.592) | 0.146±0.038 (171.226±10.671) |
| | | | **Nonconvex QCQP:** $n = 100, n_{eq} = 50, n_{ineq} = 50$ | | |
| Solver | 6.8e-14±1.6e-16 (9.9e-14±3.5e-15) | 8.9e-9±2.1e-10 (5.4e-8±1.6e-9) | – | – | 216.772±2.650 (11169.226±102.22) |
| FSNet | 9.7e-5±3.9e-6 (6.5e-6±2.9e-6) | 2.9e-6±2.4e-7 (1.1e-4±1.4e-5) | -0.668±0.650 | -19.436±1.767 (8.809±0.942) | 0.221±0.028 (303.139±25.742) |
| | | | **Nonconvex SOCP:** $n = 100, n_{eq} = 50, n_{ineq} = 50$ | | |
| Solver | 8.3e-14±1.3e-16 (1.3e-13±3.3e-15) | 9.9e-9±3.0e-10 (7.6e-8±7.6e-10) | – | – | 1216.105±7.923 (33765.581±170.102) |
| FSNet | 5.8e-5±3.5e-5 (3.4e-3±1.4e-4) | 5.2e-7±1.5e-7 (2.5e-4±1.6e-3) | -2.3e3±0.7e3 | -1.3e6±9.0e5 (67.138±3.265) | 0.406±0.051 (288.161±9.598) |

Table B.2: Test results of FSNet on 2000 instances of smooth nonconvex problems.

## B.3 Nonsmooth nonconvex problems

Table B.3 shows the detailed results of nonsmooth nonconvex problems (see Appendix A for formulations). As shown in this table, FSNet achieves near-zero constraint violations across nonsmooth nonconvex problems and achieves 152-3316× speedups in the batch setting and 7-80× speedups in the sequential setting compared to the traditional solver, which is also visualized in Figure B.5.

Notably, FSNet can attain lower-cost local solutions than the solver, as evidenced by the negative optimality gaps in Table B.3 and by the distributions of objective values in Figure B.4.

| Method | Equality Vio. Mean (Max) | Inequality Vio. Mean (Max) | Optimality Gap (%) Mean | Optimality Gap (%) Min (Max) | Runtime (s) Batch (sequential) |
|---|---|---|---|---|---|
| | | | **Nonsmooth Nonconvex QP:** $n = 100, n_{eq} = 100, n_{ineq} = 100$ | | |
| Solver | 5.4e-14±8.2e-17 (7.9e-14±6.9e-16) | 5.3e-10±5.6e-12 (1.2e-8±2.4e-10) | – | – | 19.161±0.566 (1120.035±8.607) |
| FSNet | 9.0e-5±3.8e-6 (1.5e-3±1.7e-4) | 6.3e-7±3.2e-8 (4.2e-5±6.7e-6) | 0.109±8.5e-3 | -1.081±0.275 (8.128±0.593) | 0.126±0.024 (156.553±4.761) |
| | | | **Nonconvex QCQP:** $n = 100, n_{eq} = 50, n_{ineq} = 50$ | | |
| Solver | 6.8e-14±2.4e-16 (1.0e-13±1.4e-15) | 4.9e-9±3.7e-11 (4.5e-8±3.5e-9) | – | – | 288.768±0.329 (15092.158±6116.515) |
| FSNet | 9.1e-5±4.8e-6 (6.5e-6±2.9e-6) | 2.9e-6±1.69e-7 (1.1e-4±1.4e-5) | -3.437±0.661 | -27.299±2.573 (9.466±0.919) | 0.539±0.018 (247.819±37.821) |
| | | | **Nonsmooth Nonconvex SOCP:** $n = 100, n_{eq} = 50, n_{ineq} = 50$ | | |
| Solver | 7.8e-14±1.3e-16 (1.2e-13±2.5e-15) | 3.4e-9±1.5e-10 (5.9e-8±2.7e-9) | – | – | 1363.465±5.109 (21642.764±81.569) |
| FSNet | 6.2e-5±9.4e-6 (2.5e-3±5.2e-4) | 4.2e-7±1.2e-7 (6.1e-5±6.0e-6) | -1.2e3±9.43e2 | -9.6e5±7.6e5 (658.5±831.8) | 0.411±0.019 (267.389±6.987) |

Table B.3: Test results of FSNet on 2000 instances of nonsmooth nonconvex problems.

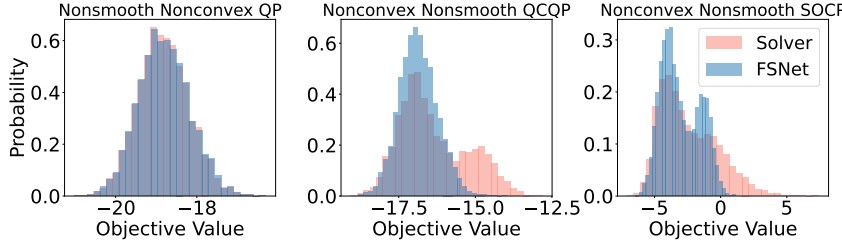

Figure B.4: Distribution of the objective values in nonsmooth nonconvex problems.

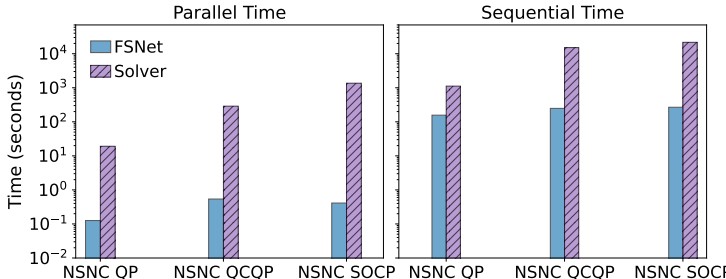

Figure B.5: Computational time on 2000 instances of nonsmooth nonconvex (NSNC) problems.

## B.4 FSNet with truncated unrolled differentiation

This section evaluates the impact of the truncation depth on the performance of FSNet (see Section 5). We set the maximum number of feasibility-seeking iterations to 50, and vary the truncation depth from 0 to 50. The results on the smooth convex problems are shown in Table B.4 and Figure B.6.

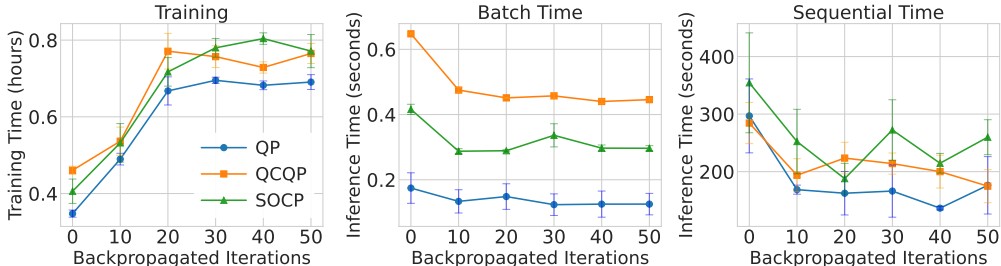

Figure B.6: Training time and batch inference time of FSNet with different truncation depths.

When $K' = 0$, no iterations of the feasibility-seeking procedure are included in the computational graph, so it acts as a non-differentiable pass-through layer. This yields the smallest graph and therefore the shortest training time, but incurs a large optimality gap, indicating the network cannot learn the true solutions of Problem (1). As $K'$ increases, the training is successful with small optimality gaps (and near-zero constraint violations), yet training time grows due to the larger graph and additional backpropagation cost (Figure B.6). Importantly, for $K'$ between 10 and 50, both total constraint violation and optimality gap remain essentially unchanged. This means that a modest value of $K'$ is sufficient to render the bias caused by the gradient truncation negligible, which agrees with our analysis in Appendix D.2. This suggests a practical tip that, rather than fully gradient-tracking all iterations of the feasibility-seeking procedure, we only need to track a few of them while maintaining high performance for FSNet and reducing the training time.

Since truncation affects only the backward pass during training, inference times are virtually identical for all choices of $K' \neq 0$.

| Tracked Iter. $K'$ | Total vio. Mean (Max) | Optimality Gap (%) Mean (Max) | Runtime (s) Batch (Sequential) | Training time (h) |
|---|---|---|---|---|
| **Convex QP:** $n = 100, n_{\text{eq}} = 100, n_{\text{ineq}} = 100$ | | | | |
| 0 | 4.4e-5±1e-6 (4.2e-4±4.2e-5) | 14.143±0.024 (16.314±0.148) | 0.174±0.046 (296.889±64.220) | 0.348±0.009 |
| 10 | 7.5e-5±6.2e-6 (6.1e-4±1e-4) | 0.015±0.001 (0.050±0.005) | 0.134±0.036 (168.696±7.896) | 0.490±0.015 |
| 20 | 6.8e-5±8.3e-6 (5.5e-4±4.6e-5) | 0.011±0.001 (0.034±0.002) | 0.149±0.039 (162.488±37.734) | 0.668±0.037 |
| 30 | 6.6e-5±5.5e-6 (9.2e-4±1.9e-4) | 0.015±0.002 (0.046±0.008) | 0.124±0.033 (166.242±45.394) | 0.695±0.008 |
| 40 | 6.6e-5±5.5e-6 (9.2e-4±1.9e-4) | 0.015±0.002 (0.046±0.008) | 0.125±0.040 (136.559±3.089) | 0.682±0.011 |
| 50 | 6.6e-5±5.5e-6 (9.2e-4±1.9e-4) | 0.015±0.002 (0.046±0.008) | 0.125±0.033 (176.246±49.894) | 0.691±0.019 |
| **Convex QCQP:** $n = 100, n_{\text{eq}} = 50, n_{\text{ineq}} = 50$ | | | | |
| 0 | 7.2e-5±4.5e-7 (7.9e-4±9.7e-5) | 37.174±0.148 (45.584±0.395) | 0.648±0.003 (284.443±35.772) | 0.461±0.011 |
| 10 | 6.2e-5±2.8e-6 (8.1e-4±6.6e-5) | 0.032±0.002 (0.405±0.062) | 0.475±0.008 (193.722±28.776) | 0.536±0.037 |
| 20 | 6.2e-5±8.0e-6 (1.1e-03±1.8e-4) | 0.032±0.002 (0.411±0.052) | 0.451±0.002 (223.545±27.455) | 0.771±0.047 |
| 30 | 6.2e-5±5.8e-6 (2.1e-03±5.7e-4) | 0.035±0.003 (0.432±0.070) | 0.457±0.009 (213.981±18.431) | 0.757±0.028 |
| 40 | 6.2e-5±5.8e-6 (2.1e-03±5.7e-4) | 0.035±0.003 (0.432±0.070) | 0.440±0.006 (200.018±28.512) | 0.729±0.015 |
| 50 | 6.2e-5±5.8e-6 (2.1e-03±5.7e-4) | 0.035±0.003 (0.432±0.070) | 0.446±0.003 (174.800±28.764) | 0.765±0.026 |
| **Convex SOCP:** $n = 100, n_{\text{eq}} = 50, n_{\text{ineq}} = 50$ | | | | |
| 0 | 4.8e-5±2e-5 (4.7e-4±7.4e-5) | 181.38±0.092 (194.957±0.896) | 0.416±0.016 (354.401±86.816) | 0.406±0.032 |
| 10 | 6.6e-5±7.3e-6 (6.2e-4±6.3e-5) | 0.136±0.010 (1.006±0.143) | 0.288±0.008 (252.299±56.041) | 0.532±0.050 |
| 20 | 6.2e-5±2.2e-6 (5.6e-4±2.2e-5) | 0.186±0.020 (1.914±0.439) | 0.289±0.003 (187.842±26.580) | 0.717±0.037 |
| 30 | 6.3e-5±6.1e-6 (6.7e-4±9.9e-5) | 0.159±0.005 (1.889±0.376) | 0.336±0.036 (272.440±52.514) | 0.780±0.024 |
| 40 | 6.3e-5±6.1e-6 (6.7e-4±9.9e-5) | 0.159±0.005 (1.889±0.376) | 0.297±0.010 (214.608±17.089) | 0.804±0.015 |
| 50 | 6.3e-5±6.1e-6 (6.7e-4±9.9e-5) | 0.159±0.005 (1.889±0.376) | 0.297±0.008 (259.888±30.394) | 0.771±0.043 |

Table B.4: Numerical results on 2000 instances of smooth convex problems with varying numbers of tracked iterations.

## B.5  FSNet with different values of $\rho$

This section investigates the impact of the value of penalty weight $\rho$ on the performance of FSNet (see Section 3.1). Table B.5 shows that the constraint violations are near-zero for all choices of $\rho$, since the feasibility-seeking step always tries to enforce the solution's feasibility regardless of the NN's prediction. This is an advantage of FSNet, as the feasibility is guaranteed by a well-designed feasibility-seeking step, rather than being left to the NN alone. While the optimality gaps are similar across different $\rho$, both training and inference times decrease as $\rho$ increases (Figure B.7). Actually, a larger penalty weight strongly penalizes the distance between the solutions before and after the feasibility-seeking step, driving the NN to produce outputs that are closer to the feasible set. This reduces the number of iterations required by the feasibility-seeking step, leading to lower computational time for the entire process. For example, setting $\rho = 50$ makes the training 1.4-1.6× faster and inference 1.3-1.7× faster compared to $\rho = 0$. However, excessively large values of $\rho$ may cause the optimization to overemphasize feasibility at the expense of optimality, potentially slowing convergence of the network parameters.

| $\rho$ | Total vio. Mean (Max) | Optimality Gap (%) Mean (Max) | Runtime (s) Batch (Sequential) | Training time (h) |
|---|---|---|---|---|
| **Convex QP:** $n = 100, n_{\text{eq}} = 100, n_{\text{ineq}} = 100$ | | | | |
| 0 | 5.8e-05±6.2e-06 (5.8e-04±1.0e-04) | 0.008±0.001 (0.029±0.003) | 0.183±0.055 (261.704±13.474) | 1.075±0.027 |
| 0.5 | 6.5e-05±1.0e-05 (9.5e-04±2.4e-04) | 0.009±0.001 (0.031±0.004) | 0.150±0.034 (236.686±50.218) | 0.827±0.076 |
| 2.5 | 6.6e-05±8.6e-06 (7.8e-04±1.4e-04) | 0.011±0.002 (0.041±0.004) | 0.183±0.014 (196.120±47.639) | 0.747±0.044 |
| 5.0 | 6.6e-05±5.5e-06 (9.2e-04±1.9e-04) | 0.015±0.002 (0.046±0.008) | 0.142±0.027 (186.968±34.144) | 0.687±0.020 |
| 10.0 | 7.1e-05±6.9e-06 (9.6e-04±2.0e-04) | 0.014±0.001 (0.042±0.004) | 0.117±0.029 (152.014±22.923) | 0.676±0.022 |
| 20.0 | 6.7e-05±2.4e-06 (1.2e-03±2.5e-04) | 0.011±0.002 (0.042±0.003) | 0.133±0.050 (165.496±23.169) | 0.649±0.013 |
| 50.0 | 5.9e-05±1.1e-05 (9.3e-04±2.5e-04) | 0.018±0.006 (0.074±0.016) | 0.104±0.013 (156.569±53.576) | 0.661±0.012 |
| **Convex QCQP:** $n = 100, n_{\text{eq}} = 50, n_{\text{ineq}} = 50$ | | | | |
| 0 | 7.2e-05±2.7e-06 (7.3e-04±1.6e-04) | 0.023±0.002 (0.315±0.064) | 0.576±0.004 (257.309±15.105) | 1.039±0.037 |
| 0.5 | 5.7e-05±1.0e-05 (1.0e-03±1.5e-04) | 0.028±0.002 (0.412±0.075) | 0.487±0.015 (216.307±32.152) | 0.811±0.010 |
| 2.5 | 7.1e-05±7.8e-06 (1.6e-03±3.6e-04) | 0.030±0.001 (0.411±0.073) | 0.464±0.004 (187.807±11.011) | 0.771±0.024 |
| 5.0 | 6.2e-05±5.8e-06 (2.1e-03±5.7e-04) | 0.035±0.003 (0.432±0.070) | 0.451±0.008 (180.282±32.059) | 0.743±0.014 |
| 10.0 | 6.1e-05±6.6e-06 (1.5e-03±1.2e-04) | 0.038±0.003 (0.417±0.058) | 0.450±0.014 (222.192±56.348) | 0.715±0.006 |
| 20.0 | 5.8e-05±6.8e-06 (2.4e-03±6.9e-04) | 0.041±0.007 (0.466±0.081) | 0.437±0.009 (240.089±33.913) | 0.762±0.052 |
| 50.0 | 5.3e-05±1.5e-06 (2.6e-03±7.5e-04) | 0.049±0.016 (0.486±0.088) | 0.447±0.015 (225.130±36.618) | 0.716±0.031 |
| **Convex SOCP:** $n = 100, n_{\text{eq}} = 50, n_{\text{ineq}} = 50$ | | | | |
| 0 | 5.3e-05±8.2e-06 (7.1e-04±1.2e-04) | 0.356±0.097 (1.490±0.278) | 0.360±0.012 (310.059±71.106) | 0.976±0.044 |
| 0.5 | 7.2e-05±8.3e-06 (6.6e-04±7.6e-05) | 0.180±0.012 (2.347±0.514) | 0.325±0.017 (237.029±72.636) | 0.921±0.014 |
| 2.5 | 6.6e-05±9.2e-06 (7.0e-04±1.1e-04) | 0.174±0.019 (0.978±0.109) | 0.294±0.007 (206.250±15.332) | 0.797±0.028 |
| 5.0 | 6.3e-05±6.1e-06 (6.7e-04±9.9e-05) | 0.159±0.005 (1.889±0.376) | 0.307±0.014 (281.764±18.464) | 0.837±0.005 |
| 10.0 | 7.9e-05±7.0e-07 (7.7e-04±8.2e-05) | 0.174±0.020 (1.365±0.251) | 0.291±0.011 (198.642±15.576) | 0.725±0.011 |
| 20.0 | 6.1e-05±4.1e-06 (9.5e-04±1.2e-04) | 0.184±0.017 (0.994±0.094) | 0.287±0.013 (197.955±33.984) | 0.714±0.041 |
| 50.0 | 6.3e-05±9.1e-06 (7.4e-04±9.2e-05) | 0.190±0.016 (1.093±0.192) | 0.269±0.015 (172.091±41.039) | 0.694±0.027 |

Table B.5: Numerical results of FSNet on 2000 instances of smooth convex problems with varying $\rho$.

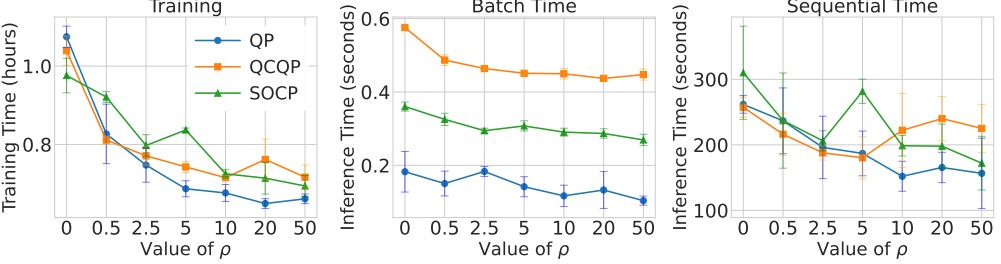

Figure B.7: Training time and batch inference time of FSNet with different values of $\rho$.

## B.6 AC Optimal Power Flow

We evaluate FSNet in solving the AC optimal power flow (ACOPF) problem in power systems. We use the formulation and dataset in [46]. In the following experiments, we use 10000 samples for each grid size, split into 8000 for training and 2000 for testing. The parameters $x$ represent the load demands, and the prediction $y_\theta(x)$ of the neural network is the voltage magnitudes and angles of all power system buses. The active and reactive power generations can then be computed from the voltage magnitudes and phase angles via the power balance equations. In particular, the power output of the generator at bus $i$ is given by

$$p_i^{\mathrm{g}} = p_i^{\mathrm{d}} + g_i^{\mathrm{s}} v_i^2 + \sum_{e \in \mathcal{E}_i} p_e^{\mathrm{f}} + \sum_{e \in \mathcal{E}_i^R} p_e^{\mathrm{t}},$$

$$q_i^{\mathrm{g}} = q_i^{\mathrm{d}} - b_i^{\mathrm{s}} v_i^2 + \sum_{e \in \mathcal{E}_i} q_e^{\mathrm{f}} + \sum_{e \in \mathcal{E}_i^R} q_e^{\mathrm{t}},$$

$$p_e^{\mathrm{f}} = g_e^{\mathrm{tt}} v_i^2 + g_e^{\mathrm{ft}} v_i v_j \cos(\theta_i - \theta_j) + b_e^{\mathrm{ft}} v_i v_j \sin(\theta_i - \theta_j), \qquad \forall e = (i,j) \in \mathcal{E},$$

$$q_e^{\mathrm{f}} = -b_e^{\mathrm{tt}} v_i^2 - b_e^{\mathrm{ft}} v_i v_j \cos(\theta_i - \theta_j) + g_e^{\mathrm{ft}} v_i v_j \sin(\theta_i - \theta_j), \qquad \forall e = (i,j) \in \mathcal{E},$$

$$p_e^{\mathrm{t}} = g_e^{\mathrm{tt}} v_j^2 + g_e^{\mathrm{tf}} v_i v_j \cos(\theta_i - \theta_j) - b_e^{\mathrm{tf}} v_i v_j \sin(\theta_i - \theta_j), \qquad \forall e = (i,j) \in \mathcal{E},$$

$$q_e^{\mathrm{t}} = -b_e^{\mathrm{tt}} v_j^2 - b_e^{\mathrm{tf}} v_i v_j \cos(\theta_i - \theta_j) - g_e^{\mathrm{tf}} v_i v_j \sin(\theta_i - \theta_j), \qquad \forall e = (i,j) \in \mathcal{E}.$$

In addition, the Levenberg-Marquardt method is employed for the feasibility-seeking step.

The results for the IEEE 30-bus and 118-bus systems under three random seeds are presented in Table B.6. FSNet achieves small (numerically zero) equality and inequality constraint violations, and significantly outperforms the Penalty method. Moreover, the optimality gap remains below $1\%$ in both test cases. In terms of runtime in the sequential setting, FSNet significantly outperforms IPOPT, achieving $3\times$ speedups on the IEEE 30-bus system and $11\times$ on the IEEE 118-bus system. The Penalty method, on the other hand, despite its fast inference, suffers from high constraint violations, particularly in terms of maximum violations on the IEEE 118-bus system. Overall, this result is consistent with our previous findings on convex and nonconvex problems.

| Method | Equality Vio. Mean (Max) | Inequality Vio. Mean (Max) | Optimality Gap (%) Mean | Min (Max) | Runtime (s) Sequential |
|---|---|---|---|---|---|
| **IEEE 30-bus:** $n = 60, n_{\mathrm{eq}} = 60, n_{\mathrm{ineq}} = 248$ | | | | | |
| IPOPT Solver | 6.7e-7±0.0 (5.8e-6±0.0) | 5.4e-9±0.0 (1.1e-6±0.0) | – | – | 51.53±0.0 |
| FSNet | 1.8e-5±7.5e-6 (1.6e-3±2.1e-4) | 1.4e-8±4.7e-9 (6.7e-4±1.7e-4) | 0.81±0.21 | 0.07±1.7e-3 (3.61±0.92) | 16.43±0.13 |
| Penalty | 8.2e-4±5.3e-5 (0.013±1.5e-3) | 3.7e-7±1.2e-7 (0.026±8.5e-3) | 2.24±0.53 | -0.13±0.86 (7.09±0.65) | 0.52±0.23 |
| **IEEE 118-bus:** $n = 236, n_{\mathrm{eq}} = 236, n_{\mathrm{ineq}} = 1196$ | | | | | |
| IPOPT Solver | 1.2e-6±0.0 (2.8e-5±0.0) | 3.1e-8±0.0 (1.1e-5±0.0) | – | – | 392.83±0.0 |
| FSNet | 7.8e-5±1.1e-5 (0.047±0.010) | 2.4e-6±2.3e-7 (0.043±8.5e-3) | 0.9±0.03 | -1.67±0.92 (4.91±0.09) | 33.35 ±0.61 |
| Penalty | 2.5e-3±3.2e-4 (0.124±0.014) | 3.5e-5±9.9e-6 (0.289±0.041) | 0.49±0.14 | -4.67±0.17 (1.51±0.25) | 0.81±0.08 |

Table B.6: Test results of FSNet on 2000 instances of ACOPF problems.

## B.7 More experiments for DC3

In [13], the authors mention that unrolling more correction steps may improve the feasibility of the prediction. To evaluate this, we ran DC3 with different numbers of correction steps on 2000 test instances of smooth convex SOCP. As shown in Table B.7, increasing the number of correction steps only yields a marginal reduction in inequality violations (due to the default small step size of $10^{-7}$), while significantly increasing runtime.

Next, we evaluate the performance of DC3 with different correction step sizes. We fixed the number of maximum correction steps and changed the stepsize to obtain the results in Table B.8. Stepsizes

| Max Correction Steps | Eq. Violation | Ineq. Violation | Optimality Gap (%) | Time (s) |
|---|---|---|---|---|
| 10 | 2.30e-14 | 0.024 | 0.051 | 0.095 |
| 50 | 2.30e-14 | 0.024 | 0.051 | 0.474 |
| 100 | 2.27e-14 | 0.024 | 0.052 | 0.948 |
| 200 | 2.28e-14 | 0.023 | 0.052 | 1.896 |
| 500 | 2.28e-14 | 0.022 | 0.054 | 4.738 |
| 1000 | 2.34e-14 | 0.020 | 0.057 | 9.526 |
| 2000 | 2.28e-14 | 0.016 | 0.061 | 19.046 |
| 5000 | 2.27e-14 | 0.010 | 0.070 | 48.024 |

Table B.7: Sensitivity of DC3 performance to the maximum number of correction steps on 2000 instances of smooth convex SOCP problem, with fixed correction stepsize of $10^{-7}$.

| Correction Stepsize | Eq. Violation | Ineq. Violation | Objective Gap (%) | Time (s) |
|---|---|---|---|---|
| 1e-07 | 2.30e-14 | 0.024 | 0.051 | 0.338 |
| 1e-06 | 2.30e-14 | 0.024 | 0.052 | 0.121 |
| 1e-05 | 2.28e-14 | 0.020 | 0.056 | 0.095 |
| 1e-04 | 2.28e-14 | 0.004 | 0.081 | 0.096 |
| 1e-03 | 2.31e-14 | 0.000 | 0.130 | 0.095 |
| 1e-02 | -3.52e+12 | 2.89e+30 | 1e58 | 0.095 |

Table B.8: Sensitivity of DC3 performance to the correction stepsize on 2000 instances of smooth convex SOCP problem, with fixed maximum correction steps of 10.

up to $10^{-5}$ achieve limited violation reduction, whereas stepsizes of $10^{-4}$ and $10^{-3}$ substantially mitigate violations. However, overly big stepsize (e.g., $10^{-2}$) causes the correction procedure to diverge. Note that all experiments here are in inference time. During training, a large correction stepsize can destabilize the optimization, as noted in [13]. This demonstrates the DC3's sensitivity to the correction stepsize, whereas our feasibility-seeking step, implemented via L-BGFS with a backtracking line search, mitigates such sensitivity and delivers robust performance across diverse problems using the same hyperparameters.

# C Hyperparameters

For DC3 and Penalty methods, we adopt the hyperparameters as tuned in [13]. For the Adaptive Penalty method, we performed a brief manual sweep to identify settings yielding strong empirical performance. For FSNet, we configured the L-BFGS solver with a sufficiently large maximum-iteration budget and memory size to guarantee convergence on all the problem classes. We also evaluated several neural network sizes and learning rates to ensure sufficient expressive capacity and stable optimization for all problem classes. Table C.1 lists all hyperparameter values.

| Hyperparameter | Value |
|---|---|
| *Shared*: | |
| Learning rate | 5e-4 |
| Learning rate decay | 0.5 |
| Number of hidden layers | 4 |
| Nonlinearity | SiLU |
| Number of hidden neurons per layer | 1024 |
| Train/validation/test ratio | 0.7/0.1/0.2 |
| Minibatch size | 512 |
| Random seeds | 2025, 2027, 2029 |
| *FS*: | |
| Equality violation weight | 10 |
| Inequality violation weight | 10 |
| Max L-BFGS iterations | 50 |
| L-BFGS memory | 30 |
| Epoch | 100 |
| Learning rate decay steps | 2000 |
| *Penalty*: | |
| Equality violation weight | 50 |
| Inequality violation weight | 50 |
| Epoch | 1000 |
| Learning rate decay steps | 4000 |
| *Adaptive Penalty*: | |
| Initial equality violation weight | 30 |
| Initial inequality violation weight | 30 |
| Max equality violation weight | 500 |
| Max inequality violation weight | 500 |
| Increasing rate | 2 |
| Epoch | 1000 |
| Learning rate decay steps | 4000 |
| *DC3*: | |
| Equality violation weight | 10 |
| Inequality violation weight | 10 |
| Correction stepsize | 1e-7 |
| Max correction stepsize | 10 |
| Correction momentum | 0.5 |
| Epoch | 1000 |
| Learning rate decay steps | 4000 |

Table C.1: Hyperparameters

# D  Analysis and Proofs

## D.1  Proof of Theorem 2

**Theorem 2.** *Suppose that $\mathbb{E}_t\left[\|\tilde{\nabla}L(\theta_t)\|_2^2\right] \leq G$ for all $t$ and that the feasibility-seeking step is unrolled for $K$ steps in every forward pass. Let $\gamma$ be as defined in Theorem 1. Under Assumption 1, 2, and 3, after running the SGD for $T$ iterations with step size $\eta = \eta_0/\sqrt{T}$ for some $\eta_0 > 0$, there is at least one iteration $t \in \{0, \ldots, T-1\}$ satisfying*

$$\mathbb{E}\left[\|\nabla L(\theta_t)\|_2^2\right] \leq \mathcal{O}\left(T^{-1/2}\right), \tag{11}$$

$$\mathbb{E}\left[\|\nabla\mathcal{L}(\theta_t)\|_2\right] \leq \mathcal{O}\left(T^{-1/4} + \gamma^K\right). \tag{12}$$

*Proof.* **Gradient bound of finite-unrolling loss.**  For the first part of the theorem, we follow the analysis of the SGD for a $L$-smooth function. By the quadratic upper bound of the $L$-smooth function, we have

$$L(\theta_{t+1}) \leq L(\theta_t) + \langle \nabla L(\theta_t), \theta_{t+1} - \theta_t \rangle + \frac{L_N}{2}\|\theta_{t+1} - \theta_t\|_2^2.$$

Observing from the SGD update (10) that $\theta_{t+1} - \theta_t = -\eta\tilde{\nabla}L(\theta_t)$, taking the conditional expectation given all randomness up to time $t$, and rearranging terms, we obtain

$$\|\nabla L(\theta_t)\|_2^2 \leq \frac{1}{\eta}(L(\theta_t) - \mathbb{E}_t[L(\theta_{t+1})]) + \frac{L_N\eta}{2}\mathbb{E}_t\left[\|\tilde{\nabla}L(\theta_t)\|_2^2\right].$$

Taking the full expectation for both sides and using the assumption $\mathbb{E}_t\left[\left\|\tilde{\nabla}L(\theta_t)\right\|_2^2\right] \leq G$ yields

$$\mathbb{E}\left[\|\nabla L(\theta_t)\|_2^2\right] \leq \frac{1}{\eta}(L(\theta_t) - \mathbb{E}[L(\theta_{t+1})]) + \frac{L_N\eta G}{2}.$$

Finally, we take the average over $t \in 0, \ldots, T-1$ and use the telescoping sum to get

$$\frac{1}{T}\sum_{t=0}^{T-1}\mathbb{E}\left[\|\nabla L(\theta_t)\|_2^2\right] \leq \frac{L(\theta_0) - L^\star}{\eta T} + \frac{L_N\eta G}{2}.$$

Since $\min_{t\in\{0,\ldots,T-1\}}\|\nabla L(\theta_t)\|_2^2 \leq \frac{1}{T}\sum_{t=0}^{T-1}\|\nabla L(\theta_t)\|_2^2$, we obtain the first statement of the theorem by replacing the stepsize $\eta = \eta_0/\sqrt{T}$.

**Gradient bound of infinite-unrolling loss.**  For any $\theta$, we consider the norm of the gradient difference:

$$\|\nabla\mathcal{L}(\theta) - \nabla L(\theta)\|_2 = \left\|\frac{1}{S}\sum_{i=1}^{S}\nabla_\theta F(y_\theta(x^{(i)}), \hat{y}_\theta(x^{(i)})) - \nabla_\theta F(y_\theta(x^{(i)}), \hat{y}_\theta^K(x^{(i)}))\right\|_2$$

$$\leq \frac{1}{S}\sum_{i=1}^{S}\left\|\nabla_\theta F(y_\theta(x^{(i)}), \hat{y}_\theta(x^{(i)})) - \nabla_\theta F(y_\theta(x^{(i)}), \hat{y}_\theta^K(x^{(i)}))\right\|_2, \quad \text{(D.1)}$$

which follows from the triangle inequality. We consider one element of the sum, and let $g := \left\|\nabla_\theta F(y_\theta(x), \hat{y}_\theta(x)) - \nabla_\theta F(y_\theta(x), \hat{y}_\theta^K(x))\right\|$. Then, via the chain rule, we have

$$g = \left\|(J_\theta y_\theta(x))^\top \nabla_y F(y_\theta(x), \hat{y}_\theta(x)) + (J_\theta y_\theta(x))^\top (J_y\mathsf{FS}(y_\theta(x); x))^\top \nabla_{\hat{y}} F(y_\theta(x), \hat{y}_\theta(x))\right.$$
$$\left. - (J_\theta y_\theta(x))^\top \nabla_y F(y_\theta(x), \hat{y}_\theta^K(x)) - (J_\theta y_\theta(x))^\top (J_y\mathsf{FS}^K(y_\theta(x); x))^\top \nabla_{\hat{y}} F(y_\theta(x), \hat{y}_\theta^K(x))\right\|_2$$
$$\leq B_N\left\|\nabla_y F(y_\theta(x), \hat{y}_\theta(x)) - \nabla_y F(y_\theta(x), \hat{y}_\theta^K(x))\right\|_2$$
$$+ B_N\left\|(J_y\mathsf{FS}(y_\theta(x); x))^\top \nabla_{\hat{y}} F(y_\theta(x), \hat{y}_\theta(x)) - (J_y\mathsf{FS}^K(y_\theta(x); x))^\top \nabla_{\hat{y}} F(y_\theta(x), \hat{y}_\theta^K(x))\right\|_2.$$

where the inequality follows from the bounded Jacobian in Assumption 2.

The first term becomes (by expansion of the gradient terms):

$$\left\|\nabla_y F(y_\theta(x), \hat{y}_\theta(x)) - \nabla_y F(y_\theta(x), \hat{y}_\theta^K(x))\right\|_2 = \rho\left\|\hat{y}_\theta(x) - \hat{y}_\theta^K(x)\right\|_2.$$

The second term:

$$\left\|(J_y\mathsf{FS}(y_\theta(x); x))^\top \nabla_{\hat{y}} F(y_\theta(x), \hat{y}_\theta(x)) - (J_y\mathsf{FS}^K(y_\theta(x); x))^\top \nabla_{\hat{y}} F(y_\theta(x), \hat{y}_\theta^K(x))\right\|_2$$

$$=\left\|(J_y\mathsf{FS}(y_\theta(x); x))^\top \nabla_{\hat{y}} F(y_\theta(x), \hat{y}_\theta(x)) - (J_y\mathsf{FS}^K(y_\theta(x); x))^\top \nabla_{\hat{y}} F(y_\theta(x), \hat{y}_\theta(x))\right.$$
$$\left. + (J_y\mathsf{FS}^K(y_\theta(x); x))^\top \nabla_{\hat{y}} F(y_\theta(x), \hat{y}_\theta(x)) - (J_y\mathsf{FS}^K(y_\theta(x); x))^\top \nabla_{\hat{y}} F(y_\theta(x), \hat{y}_\theta^K(x))\right\|_2$$

$$\leq\left\|J_y\mathsf{FS}(y_\theta(x); x) - J_y\mathsf{FS}^K(y_\theta(x); x)\right\|_2\left\|\nabla_{\hat{y}} F(y_\theta(x), \hat{y}_\theta(x))\right\|_2$$
$$+ \left\|J_y\mathsf{FS}^K(y_\theta(x); x)\right\|_2\left\|\nabla_{\hat{y}} F(y_\theta(x), \hat{y}_\theta(x)) - \nabla_{\hat{y}} F(y_\theta(x), \hat{y}_\theta^K(x))\right\|_2$$

$$\leq B_F L_{FS}\left\|\hat{y}_\theta(x) - \hat{y}_\theta^K(x)\right\|_2 + B_{FS} L_F\left\|\hat{y}_\theta(x) - \hat{y}_\theta^K(x)\right\|_2$$

$$=(B_F L_{FS} + B_{FS} L_F)\left\|\hat{y}_\theta(x) - \hat{y}_\theta^K(x)\right\|_2,$$

where the second inequality follows from the smoothness of the feasibility-seeking mapping and function $F(y_\theta(x), \hat{y}_\theta(x))$ (Assumption 2).

Let $\nu = \rho + B_F L_{FS} + B_{FS} L_F$. Thus, $g$ is upper bounded by

$$g \leq \nu\|\hat{y}_\theta^K(x) - \hat{y}_\theta(x)\|_2.$$

Using this upper bound and Theorem 1, we can rewrite the gradient difference (D.1) as follows:

$$\|\nabla\mathcal{L}(\theta) - \nabla L(\theta)\|_2 \leq \frac{1}{S}\sum_{i=1}^S \nu\left\|\hat{y}_\theta^K(x^{(i)}) - \hat{y}_\theta(x^{(i)})\right\|_2$$

$$\leq \frac{1}{S}\sum_{i=1}^S \frac{8\nu\eta_\phi L_\phi^2}{\mu_\phi^2}(\phi(y_\theta^{(i)}; x^{(i)}) - \phi(\hat{y}_\theta(x^{(i)}); x^{(i)})), \qquad \text{(D.2)}$$

where $\hat{y}_\theta(x^{(i)}) = \mathsf{FS}(y_\theta(x^{(i)}); x^{(i)})$.

Let $\Phi(\theta) := \frac{1}{S}\sum_{i=1}^S \frac{8\nu\eta_\phi L_\phi^2}{\mu_\phi^2}\gamma^{K-1}\big(\phi(y_\theta(x^{(i)}); x^{(i)}) - \phi(\hat{y}_\theta(x^{(i)}); x^{(i)})\big)$. Using the inverse triangle inequality, the inequality (D.2) leads to

$$\|\nabla\mathcal{L}(\theta)\|_2 \leq \|\nabla L(\theta)\|_2 + \Phi(\theta)\gamma^{K-1}.$$

We consider the time $t$ and use (11) to obtain

$$\mathbb{E}[\|\nabla\mathcal{L}(\theta_t)\|_2] \leq \mathbb{E}[\|\nabla L(\theta_t)\|_2] + \mathbb{E}[\Phi(\theta_t)]\gamma^{K-1}$$

$$\leq \mathcal{O}(T^{-1/4}) + \mathbb{E}[\Phi(\theta_t)]\gamma^{K-1}. \qquad \text{(D.3)}$$

Finally, we need to show that $\mathbb{E}[\Phi(\theta_t)]$ is upper bounded. We have

$$\mathbb{E}[\Phi(\theta_t)] = \frac{1}{S}\sum_{i=1}^S \frac{8\nu\eta_\phi L_\phi^2}{\mu_\phi^2}\mathbb{E}\left[\phi\big(y_\theta(x^{(i)}); x^{(i)}\big) - \phi\big(\hat{y}_\theta(x^{(i)}); x^{(i)}\big)\right]. \qquad \text{(D.4)}$$

Since Theorem 1 shows that the gradient descent converges to the minimum, we have $\phi\big(y_\theta(x^{(i)}); x^{(i)}\big) \geq \phi\big(\hat{y}_\theta(x^{(i)}); x^{(i)}\big)$. Thus, it suffices to show $\mathbb{E}[\phi(y_\theta(x^{(i)}); x^{(i)})]$ is upper bounded for any $i$. The argument proceeds as follows: We first show that the NN parameters remain bounded during training with SGD, which implies that the network output $y_\theta(x^{(i)})$ is also bounded. From there, we conclude that the expectation $\mathbb{E}[\phi(y_\theta(x^{(i)}); x^{(i)})]$ is bounded from above.

*Bound of the NN parameters:*
We first establish the bounds of the NN parameters during training with SGD, which via (10) exhibits

$$\|\theta_{t+1} - \theta_t\|_2 = \eta\|\tilde{\nabla} L(\theta_t)\|_2, \quad t = 0, \ldots, T-1.$$

Taking the sum over $j = 0, \ldots, t-1$ and using triangle inequality and the telescoping sum, we have

$$\|\theta_t - \theta_0\|_2 \leq \sum_{j=0}^{t-1}\|\theta_{j+1} - \theta_j\|_2 = \sum_{j=0}^{t-1}\eta\|\tilde{\nabla} L(\theta_j)\|_2.$$

Taking the square for both sides and using the Cauchy-Schwarz inequality yields

$$\|\theta_t - \theta_0\|_2^2 \leq \sum_{j=0}^{t-1} \eta^2 \sum_{j=0}^{t-1} \|\tilde{\nabla}L(\theta_j)\|_2^2 \leq \eta^2 T \sum_{j=0}^{t-1} \|\tilde{\nabla}L(\theta_j)\|_2^2.$$

Using the assumption of bounded variance of the stochastic gradients, we obtain

$$\mathbb{E}_t[\|\theta_t - \theta_0\|_2^2] \leq \eta^2 T^2 G.$$

By replacing $\eta = \eta_0/\sqrt{T}$ and taking a regular expectation, we finally obtain

$$\mathbb{E}[\|\theta_t - \theta_0\|_2^2] \leq \eta_0^2 T G. \tag{D.5}$$

*Bound of the NN output:*

For notational brevity, we denote $y_0 = y_{\theta_0}(x)$ and $y_t = y_{\theta_t}(x)$, which are outputs of NNs with initialized parameter $\theta_0$ and parameter $\theta_t$ at time $t$. By the mean value theorem and the bounded Jacobian $\|J_\theta y_\theta(x)\|_2 \leq B_N$ in Assumption 2, we have

$$\|y_t - y_0\|_2 \leq B_N \|\theta_t - \theta_0\|_2,$$

which, squaring both sides, leads to

$$\mathbb{E}[\|y_t - y_0\|_2^2] \leq B_N^2 \mathbb{E}[\|\theta_t - \theta_0\|_2^2] \leq B_N^2 \eta_0^2 T G. \tag{D.6}$$

*Bound of $\mathbb{E}[\phi(y_t; x)]$:*

By the $L$-smoothness of $\phi(y_t; x)$, we have the quadratic upper bound:

$$\phi(y_t; x) \leq \phi(y_0; x) + \langle \nabla_y \phi(y_0; x), y_t - y_0 \rangle + \frac{L_\phi}{2} \|y_t - y_0\|_2^2.$$

Taking the expectation on both sides and using (D.6) yields

$$\mathbb{E}[\phi(y_t; x)] \leq \phi(y_0; x) + B_N \eta \sqrt{TG} \|\nabla_y \phi(y_0; x)\|_2 + \frac{L_\phi}{2} B_N^2 \eta_0^2 T G < \infty.$$

Therefore, we have that $\mathbb{E}[\phi(y_\theta(x^{(i)}); x^{(i)})]$ is upper bounded for any $i$, which implies that $\mathbb{E}[\phi(\hat{y}_\theta(x^{(i)}); x^{(i)})]$ is similarly bounded given that $\phi(\hat{y}_\theta(x^{(i)}); x^{(i)}) \leq \phi(y_\theta(x^{(i)}); x^{(i)})$ by Theorem 1 Finally, by (D.4), there exists a constant $\kappa > 0$ such that $\mathbb{E}[\Phi(\theta_t)] \leq \kappa$ for $t = 0, \ldots, T-1$. Using this in (D.3), we complete the second statement of the theorem. □

### D.1.1 Proof of Lemma 1

**Lemma 1.** *Suppose SGD converges to $\theta^\star$, where $J(\theta^\star) := \left( (J_\theta y_{\theta^\star}(x^{(1)}))^\top \; \ldots \; (J_\theta y_{\theta^\star}(x^{(S)}))^\top \right)$ has full column rank. Then, the prediction of the NN $y_\theta(x^{(i)})$ is the stationary point of $F(y_\theta(x^{(i)}), \hat{y}_\theta(x^{(i)}))$ with $\hat{y}_\theta(x^{(i)}) = \mathsf{FS}(y_\theta(x^{(i)}); x^{(i)})$ for all $i \in \{1, \ldots, S\}$.*

*Proof.* At the stationary parameter $\theta^\star$, the gradient of the loss function vanishes:

$$0 = \nabla \mathcal{L}(\theta^\star) = \frac{1}{S} \sum_{i=1}^{S} (J_\theta y_{\theta^\star}(x^{(i)}))^\top \left( \nabla_y F(y_\theta(x^{(i)}), \hat{y}_\theta(x^{(i)})) \right.$$

$$\left. + (J_y \mathsf{FS}(y_\theta(x^{(i)}); x^{(i)}))^\top \nabla_{\hat{y}} F(y_\theta(x^{(i)}), \hat{y}_\theta(x^{(i)})) \right).$$

Let $v^{(i)} = \nabla_y F(y_\theta(x^{(i)}), \hat{y}_\theta(x^{(i)})) + (J_y \mathsf{FS}(y^{(i)}; x^{(i)}))^\top \nabla_{\hat{y}} F(y_\theta(x^{(i)}), \hat{y}_\theta(x^{(i)}))$. The stationarity results in

$$J(\theta^\star) \begin{pmatrix} v^{(1)} \\ \vdots \\ v^{(S)} \end{pmatrix} = 0.$$

Since $J(\theta^\star)$ has full column rank, this implies $(v^{(1)}, \ldots, v^{(S)})^\top = 0$. Thus, the NN output $y_{\theta^\star}(x^{(i)})$ and $\hat{y}_\theta(x^{(i)}) = \mathsf{FS}(y_\theta(x^{(i)}); x^{(i)})$ satisfy

$$v^{(i)} = \nabla_y F(y_\theta(x^{(i)}), \hat{y}_\theta(x^{(i)})) + (J_y \mathsf{FS}(y^{(i)}; x^{(i)}))^\top \nabla_{\hat{y}} F(y_\theta(x^{(i)}), \hat{y}_\theta(x^{(i)})) = 0,$$

which is the stationary condition of $F(y_\theta(x^{(i)}), \hat{y}_\theta(x^{(i)}))$. □

### D.1.2 Proof of Theorem 3

**Theorem 3.** *Given specific parameters $x$, suppose that the NN predicts $y_\tau$ which is a global minimizer of $F(y, \hat{y}; \rho_\tau) = f(\hat{y}; x) + \frac{\rho_\tau}{2}\|y - \hat{y}\|_2^2$ with $\hat{y} = \mathsf{FS}(y; x)$; that the mapping $\mathsf{FS}$ is continuous; and that $0 < \rho_\tau < \rho_{\tau+1}, \forall \tau$ and $\rho_\tau \to \infty$. Then, every limit point $(y^\star, \hat{y}^\star)$ of the sequence $\{(y_\tau, \hat{y}_\tau)\}$ admits $y^\star = \hat{y}^\star$, and $\hat{y}^\star$ is a global minimizer of the optimization problem (1).*

*Proof.* Let $\overline{y}$ be a global solution to problem (1), that is

$$f(\overline{y}; x) \le f(\hat{y}; x)$$

for all the feasible points $\hat{y}$. We also have that $\overline{y} = \mathsf{FS}(\overline{y}; x)$ given that $\overline{y}$ is already feasible. By the global optimality of $(y_\tau, \hat{y}_\tau)$, we have that $F(y_\tau, \hat{y}_\tau; \rho_\tau) \le F(\overline{y}, \overline{y}; \rho_\tau)$ which results in the inequality

$$f(\hat{y}_\tau; x) + \frac{\rho_\tau}{2}\|y_\tau - \hat{y}_\tau\|_2^2 \le f(\overline{y}; x) + \frac{\rho_\tau}{2}\|\overline{y} - \overline{y}\|_2^2 = f(\overline{y}; x). \tag{D.7}$$

By rearranging this inequality, we obtain

$$\|y_\tau - \hat{y}_\tau\|_2^2 \le \frac{2}{\rho_\tau}(f(\overline{y}; x) - f(\hat{y}_\tau; x)). \tag{D.8}$$

Suppose that $y^\star$ is a limit point of $\{y_\tau\}_{\tau \ge 0}$, so that there is an infinite subsequence $\mathcal{T}$ such that

$$\lim_{\tau \in \mathcal{T}} y_\tau = y^\star.$$

As the feasibility seeking mapping is continuous by assumption, we can define $\hat{y}^\star := \lim_{\tau \in \mathcal{T}} \mathsf{FS}(y_\tau; x)$. By the continuity, we have

$$\lim_{\tau \in \mathcal{T}} \|y_\tau - \hat{y}_\tau\|_2^2 = \lim_{\tau \in \mathcal{T}} \|y_\tau - \mathsf{FS}(y_\tau; x)\|_2^2 = \|y^\star - \hat{y}^\star\|_2^2.$$

Taking the limit as $\tau \to \infty, \tau \in \mathcal{T}$ on both sides of (D.8) yields

$$\|y^\star - \hat{y}^\star\|_2^2 = \lim_{\tau \in \mathcal{T}} \|y_\tau - \hat{y}_\tau\|_2^2 \le \lim_{\tau \in \mathcal{T}} \frac{2}{\rho_\tau}(f(\overline{y}; x) - f(\hat{y}_\tau; x)) = 0.$$

Therefore, we have that $y^\star = \hat{y}^\star$. Note that $\hat{y}^\star$ is the output of the feasibility-seeking mapping, so $y^\star$ and $\hat{y}^\star$ are feasible. Moreover, by taking the limit as $\tau \to \infty, \tau \in \mathcal{T}$ in (D.7), we have

$$f(\hat{y}^\star) \le f(\hat{y}^\star) + \lim_{\tau \in \mathcal{T}} \frac{\rho_\tau}{2}\|y_\tau - \hat{y}_\tau\|_2^2 \le f(\overline{y}; x).$$

Since $\hat{y}^\star$ is a feasible point with an objective value no larger than that of the global solution $\overline{y}$, it must itself be a global solution. $\square$

Using a similar argument, we can show that $\hat{y}^\star$ is a minimizer of Problem 1 under the same assumptions, even when $\rho = 0$. However, in practice, we often choose $\rho > 0$ for the reasons discussed in Section 3.1 and Appendix B.

### D.2 Analysis of Truncated Unrolled Differentiation

In this section, we build intuition for why truncating backpropagation remains effective in practice. Specifically, if the violation function $\phi(s, x)$ is locally strongly convex, the bias introduced by truncation decays exponentially with the truncation depth $K'$. Consequently, a relatively small $K'$ suffices to yield a high-quality gradient estimate.

**Assumption 4.** *The violation function $s \mapsto \phi(s; x)$ is twice differentiable. For $s_K \in \mathbb{R}^n$, there exists a neighborhood $\mathcal{N}$ of $s_K$ where $s \mapsto \phi(s; x)$ is $\mu'$-strongly convex. Moreover, for some $K_0 \in \mathbb{N}$, the iterates satisfy $s_k \in \mathcal{N}$ for $k \ge K_0$.*

Although the violation function involves a $\max$ operation and is not twice differentiable at the kink points of the $\max$, we can approximate the $\max$ in practice using the softplus function, which is infinitely differentiable and can approximate the $\max$ arbitrarily well. Therefore, this assumption remains reasonable in practice due to the smooth approximation provided by the softplus function. This assumption enables us to bound the bias caused by the truncation as in the following lemma.

**Lemma 2.** *Suppose Assumption 4 holds and that the feasibility-seeking procedure is unrolled with the stepsize $\eta_\phi \in (0, 1/L_\phi]$ for a total of $K$ iterations, but gradient tracking is performed only for the first $K' \in [K_0, K]$ iterations, with the remaining iterations treated as pass-through during backpropagation. Define the true Jacobian when all $K$ iterations are tracked as $J_y\mathsf{FS}^K(y_\theta(x); x)$ and the truncated Jacobian as $J_y^{\mathrm{trun}}\mathsf{FS}^K(y_\theta(x); x)$:*

$$J_y\mathsf{FS}^K(y_\theta(x); x) = \frac{\partial s_K}{\partial s_0}, \qquad J_y^{\mathrm{trun}}\mathsf{FS}^K(y_\theta(x); x) = \frac{\partial s_{K'}}{\partial s_0}$$

*Then, the bias caused by the truncation satisfies*

$$\left\| J_y\mathsf{FS}^K(y_\theta(x); x) - J_y^{\mathrm{trun}}\mathsf{FS}^K(y_\theta(x); x) \right\|_2 \le C(1 - \delta^{K-K'})\delta^{K'}$$

*where $\delta = 1 - \eta_\phi\mu' \in [0, 1)$ and $C$ is a finite positive constant.*

Since the truncation occurs during the backward pass, it also perturbs the stochastic gradient. We denote the resulting truncated gradient estimate by $\tilde{\nabla}^{\mathrm{trun}}L(\theta_t)$. The next lemma quantifies this effect.

**Lemma 3.** *For gradient estimate without truncation, suppose $\mathbb{E}_t\left[\|\tilde{\nabla}L(\theta_t)\|_2^2\right] \le G$ and $\mathbb{E}_t[\tilde{\nabla}L(\theta_t)] = \nabla L(\theta_t)$ for all $t$. Then, when the gradient truncation is applied, under Assumptions in Lemma 2, the truncated gradient estimate satisfies*

$$\mathbb{E}_t\left[\|\tilde{\nabla}^{\mathrm{trun}}L(\theta_t)\|_2^2\right] \le G + \sigma_{K'},$$

$$\mathbb{E}_t\left[\tilde{\nabla}^{\mathrm{trun}}L(\theta_t)\right] = \nabla L(\theta_t) + \varepsilon_{K'}.$$

*where $\sigma_{K'} = \mathcal{O}(\delta^{K'})$ and $\varepsilon_{K'} \in \mathbb{R}^{n_\theta}$ with $\|\varepsilon_{K'}\| = \mathcal{O}(\delta^{K'})$.*

This lemma shows that the truncation introduces bounded biases to the variance and expectation of the gradient estimate, yet these biases decay exponentially fast in the truncation depth $K'$. This qualification allows us to understand the behavior of the SGD when the truncation is applied.

**Theorem 4.** *Suppose Assumption 1, 2, 3, and the assumptions of Lemma 3 hold. Then, after running the SGD for $T$ iterations with step size $\eta = \eta_0/\sqrt{T}$ with $\eta_0 > 0$, there is at least one iteration $t \in \{0, \dots, T-1\}$ satisfying*

$$\mathbb{E}\left[\|\nabla L(\theta_t)\|_2^2\right] \le \mathcal{O}\left(T^{-1/2} + \delta^{K'}\right), \tag{D.9}$$

$$\mathbb{E}[\|\nabla\mathcal{L}(\theta_t)\|_2] \le \mathcal{O}\left(T^{-1/4} + \gamma^K + \delta^{K'}\right), \tag{D.10}$$

*where $\gamma$ and $\delta$ are defined as in Theorem 4 and Lemma 2.*

This theorem characterizes how the bias caused by truncating the gradient tracking in the feasibility-seeking procedure propagates to the convergence of SGD. We see that the upper bounds of the expected gradient norms contain bias terms of order $\mathcal{O}(\gamma^K)$ and $\mathcal{O}(\delta^{K'})$. Since $0 < \gamma, \delta \le 1$, even modest values of $K$ and $K'$ suffice to render these biases negligible in practice. We now present the proofs of the above lemmas and theorems.

### D.2.1 Proof of Lemma 2

Lemma 2 is an immediate result of the following lemma.

**Lemma 4.** *Let $\varphi : \mathbb{R}^d \to \mathbb{R}$ be twice differentiable, $L$-smooth, satisfy the PL condition with constant $\mu$. Suppose there exists a neighborhood $\mathcal{N}$ of $s_K$ on which $\varphi$ is strongly convex, i.e., $\nabla^2\varphi(s) \succeq \mu'I, \forall s \in \mathcal{N}$ with $\mu' > 0$. Define*

$$J^{\mathrm{true}} = \frac{\partial s_K}{\partial s_0} = \prod_{k=0}^{K-1}(I - \eta\nabla^2\varphi(s_k)), \quad J^{\mathrm{trun}} = \frac{\partial s_{K'}}{\partial s_0} = \prod_{k=0}^{K'-1}(I - \eta\nabla^2\varphi(s_k)).$$

*If $s_k \in \mathcal{N}$ for all $k \ge K_0$ and stepsize of gradient descent is chosen $\eta \in (0, 1/L]$, then for any $K' \in [K_0, K]$, the error matrix $E = J^{\mathrm{true}} - J^{\mathrm{trun}}$ satisfies*

$$\|E\|_2 \le C(1 - \delta^{K-K'})\delta^{K'},$$

*where $\delta = 1 - \eta\mu' \in [0, 1)$ and $C = \frac{L}{\mu'}\delta^{-K_0}(1 + \eta L)^{K_0}$.*

*Proof.* Define $J_k = \frac{\partial s_{k+1}}{\partial s_k} = I - \eta \nabla^2 \varphi(s_k)$ and rewrite the norm of error matrix $E$ as

$$
\begin{aligned}
\|E\|_2 &= \|(J_{K-1} \dots J_{K'} - I)(J_{K'-1} \dots J_0)\|_2 \\
&\leq \|J_{K-1} \dots J_{K'} - I\|_2 \|J_{K'-1} \dots J_0\|_2.
\end{aligned}
\tag{D.11}
$$

**Step 1:** $0 \leq k < K'$.
For $k < K_0$, the $L$-smoothness gives $-LI \preceq \nabla^2 \varphi(s_k) \preceq LI$, so

$$
\|J_k\|_2 = \|I - \eta \nabla^2 \varphi(s_k)\|_2 \leq 1 + \eta L.
$$

For $K_0 \leq k < K'$, strong convexity implies $\mu' I \preceq \nabla^2 \varphi(s_k) \preceq LI$, and then we obtain

$$
\|J_k\|_2 \leq 1 - \eta \mu'.
$$

Let $\delta := 1 - \eta \mu' \in [0, 1)$. Then, we use the sub-multiplicativity property of the spectral norm to obtain

$$
\|(J_{K'-1} \dots J_0)\|_2 \leq \|(J_{K'-1} \dots J_{K_0})\|_2 \|(J_{K_0-1} \dots J_0)\|_2 \leq \delta^{K'-K_0}(1 + \eta L)^{K_0}.
\tag{D.12}
$$

**Step 2:** $K' \leq k < K$.
Use the telescoping sum for $\|(J_{K-1} \dots J_{K'} - I)\|_2$:

$$
\begin{aligned}
\|J_{K-1} \dots J_{K'} - I\|_2 &= \left\| \sum_{k=K'}^{K-1} \left( \prod_{j=k+1}^{K-1} J_j \right) (J_k - I) \right\|_2 \\
&\leq \sum_{k=K'}^{K-1} \left\| \prod_{j=k+1}^{K-1} J_j \right\|_2 \|J_k - I\|_2.
\end{aligned}
$$

From the $L$-smoothness we have $\|J_k - I\|_2 = \|\eta \nabla^2 \varphi(s_k)\|_2 \leq \eta L$. For $k \geq K' > K_0$, by assumption, we are in the neighborhood $\mathcal{N}$ where $\varphi$ is locally strongly convex, and thus we obtain

$$
\left\| \prod_{j=k+1}^{K-1} J_j \right\|_2 \leq \delta^{K-k-1},
$$

which results in

$$
\|J_{K-1} \dots J_{K'} - I\|_2 \leq \eta L \sum_{k=K'}^{K-1} \delta^{K-k-1}.
$$

We change the index of the summation to $i = K - k - 1$. When $k = K'$, $i = K - K' - 1$. When $k = K - 1$, $i = 0$. The sum becomes a finite geometric series with $0 \leq \delta < 1$:

$$
\begin{aligned}
\|J_{K-1} \dots J_{K'} - I\|_2 &\leq \eta L \sum_{i=0}^{K-K'-1} \delta^i \\
&= \eta L \frac{1 - \delta^{K-K'}}{1 - \delta} \\
&= \frac{L}{\mu'}(1 - \delta^{K-K'}).
\end{aligned}
\tag{D.13}
$$

**Step 3:** Combining the bounds
Substituting (D.12) and (D.13) into the inequality of error matrix (D.11) yields

$$
\|E\|_2 \leq \frac{L}{\mu'}(1 - \delta^{K-K'})\delta^{K'-K_0}(1 + \eta L)^{K_0}.
$$

Define the constant $C = \frac{L}{\mu'}\delta^{-K_0}(1 + \eta L)^{K_0}$. We can rewrite the bound as follows:

$$
\|E\|_2 \leq C(1 - \delta^{K-K'})\delta^{K'}.
$$

$\square$

### D.2.2 Proof of Lemma 3

*Proof.* For a minibatch with $D$ samples, we have

$$\left\| \tilde{\nabla} L(\theta_t) - \tilde{\nabla}^{\mathrm{trun}} L(\theta_t) \right\|_2$$

$$= \left\| \frac{1}{D} \sum_{i=1}^{D} (J_\theta y_\theta(x^{(i)}))^\top \nabla_y F(y_\theta(x^{(i)}), \hat{y}_\theta^K(x^{(i)})) \right.$$
$$+ (J_\theta y_\theta(x^{(i)}))^\top (J_y \mathsf{FS}^K(y_\theta(x)^{(i)}; x^{(i)}))^\top \nabla_{\hat{y}} F(y_\theta(x^{(i)}), \hat{y}_\theta^K(x^{(i)}))$$
$$- (J_\theta y_\theta(x^{(i)}))^\top \nabla_y F(y_\theta(x^{(i)}), \hat{y}_\theta^K(x^{(i)}))$$
$$\left. - (J_\theta y_\theta(x^{(i)}))^\top (J_y^{\mathrm{trun}} \mathsf{FS}^K(y_\theta(x)^{(i)}; x^{(i)}))^\top \nabla_{\hat{y}} F(y_\theta(x^{(i)}), \hat{y}_\theta^K(x^{(i)})) \right\|_2$$

$$= \left\| \frac{1}{D} \sum_{i=1}^{D} (J_\theta y_\theta(x^{(i)}))^\top (J_y \mathsf{FS}^K(y_\theta(x)^{(i)}; x^{(i)}) \right.$$
$$\left. - J_y^{\mathrm{trun}} \mathsf{FS}^K(y_\theta(x)^{(i)}; x^{(i)}))^\top \nabla_{\hat{y}} F(y_\theta(x)^{(i)}, \hat{y}_\theta^K(x^{(i)})) \right\|_2$$

$$\leq \frac{1}{D} \sum_{i=1}^{D} B_N B_F C (1 - \delta^{K-K'}) \delta^{K'} \qquad \text{(by Lemma 2)}$$

$$= B_N B_F C (1 - \delta^{K-K'}) \delta^{K'}. \tag{D.14}$$

Then, by triangle inequality, we have the following bound

$$\mathbb{E}_t \left[ \left\| \tilde{\nabla}^{\mathrm{trun}} L(\theta_t) \right\|_2^2 \right] \leq \mathbb{E}_t \left[ \left( \left\| \tilde{\nabla}^{\mathrm{trun}} L(\theta_t) - \tilde{\nabla} L(\theta_t) \right\|_2 + \left\| \tilde{\nabla} L(\theta_t) \right\|_2 \right)^2 \right]$$
$$\leq B_N B_F C (1 - \delta^{K-K'})^2 \delta^{2K'} + G + 2 B_N B_F C \sqrt{G} (1 - \delta^{K-K'}) \delta^{K'}$$
$$= G + \mathcal{O}(\delta^{K'}).$$

In addition, by (D.14), we have that

$$\left\| \tilde{\nabla} L(\theta_t) - \tilde{\nabla}^{\mathrm{trun}} L(\theta_t) \right\|_\infty \leq \left\| \tilde{\nabla} L(\theta_t) - \tilde{\nabla}^{\mathrm{trun}} L(\theta_t) \right\|_2 \leq B_N B_F C (1 - \delta^{K-K'}) \delta^{K'} := \upsilon_{K'},$$

which results in

$$\tilde{\nabla} L(\theta_t) - \upsilon_{K'} \mathbf{1} \leq \tilde{\nabla}^{\mathrm{trun}} L(\theta_t) \leq \tilde{\nabla} L(\theta_t) + \upsilon_{K'} \mathbf{1} \quad \text{(element-wise)}.$$

There exists $\varepsilon_{K'} \in \mathbb{R}^{n_\theta}$ such that $\|\varepsilon_{K'}\|_2 \leq \upsilon_{K'} \sqrt{n_\theta}$ that satisfies

$$\mathbb{E}_t \left[ \tilde{\nabla}^{\mathrm{trun}} L(\theta_t) \right] = \mathbb{E}_t \left[ \tilde{\nabla} L(\theta_t) \right] + \varepsilon_{K'} = \nabla L(\theta_t) + \varepsilon_{K'}.$$

Since $\upsilon_{K'} = \mathcal{O}(\delta^{K'})$, the proof is complete. $\qquad \square$

### D.2.3 Proof of Theorem 4

*Proof.* **Gradient bound for finite-unrolling loss with truncated gradient.** By the quadratic upper bound of the $L$-smooth function and bounded variance of the stochastic gradient (Lemma 3):

$$\mathbb{E}_t[L(\theta_{t+1})] \leq L(\theta_t) - \eta \langle \nabla L(\theta_t), \mathbb{E}_t[\tilde{\nabla}^{\mathrm{trun}} L(\theta_t)] \rangle + \frac{L_N \eta^2}{2} \mathbb{E}_t \left[ \|\tilde{\nabla}^{\mathrm{trun}} L(\theta_t)\|_2^2 \right]$$

$$= L(\theta_t) - \eta \langle \nabla L(\theta_t), \nabla L(\theta_t) + \varepsilon_{K'} \rangle + \frac{L_N \eta^2}{2} (G + \sigma_{K'}). \tag{D.15}$$

We observe that

$$\langle \nabla L(\theta_t), \nabla L(\theta_t) + \varepsilon_{K'} \rangle = \|\nabla L(\theta_t)\|_2^2 + \langle \nabla L(\theta_t), \varepsilon_{K'} \rangle + \frac{1}{2}\|\varepsilon_{K'}\|_2^2 - \frac{1}{2}\|\varepsilon_{K'}\|_2^2$$

$$= \frac{1}{2}\|\nabla L(\theta_t)\|_2^2 - \frac{1}{2}\|\varepsilon_{K'}\|_2^2 + \frac{1}{2}\left(\|\nabla L(\theta_t)\|_2^2 + 2\langle \nabla L(\theta_t), \varepsilon_{K'} \rangle + \|\varepsilon_{K'}\|_2^2\right)$$

$$= \frac{1}{2}\|\nabla L(\theta_t)\|_2^2 - \frac{1}{2}\|\varepsilon_{K'}\|_2^2 + \frac{1}{2}\|\nabla L(\theta_t) + \varepsilon_{K'}\|_2^2$$

$$\geq \frac{1}{2}\|\nabla L(\theta_t)\|_2^2 - \frac{1}{2}\|\varepsilon_{K'}\|_2^2.$$

Using this in (D.15) and rearranging terms, we obtain

$$\|\nabla L(\theta_t)\|_2^2 \leq \frac{2}{\eta}(L(\theta_t) - \mathbb{E}_t[L(\theta_{t+1})]) + \|\varepsilon_{K'}\|_2^2 + L_N\eta(G + \sigma_{K'}).$$

Taking the expectation for both sides and the average over $T$ steps, we obtain

$$\frac{1}{T}\sum_{t=0}^{T-1}\mathbb{E}[\|\nabla L(\theta_t)\|_2^2] \leq \frac{2}{\eta T}\mathbb{E}\left[\sum_{t=0}^{T-1} L(\theta_t) - L(\theta_{t+1})\right] + \|\varepsilon_{K'}\|_2^2 + L\eta(G + \sigma_{K'})$$

$$\leq \frac{2}{\eta T}\left(L(\theta_0) - L^\star\right) + \|\varepsilon_{K'}\|_2^2 + L_N\eta(G + \sigma_{K'}).$$

Since $\min_{t\in\{0,\ldots,T-1\}}\|\nabla L(\theta_t)\|_2^2 \leq \frac{1}{T}\sum_{t=0}^{T-1}\|\nabla L(\theta_t)\|_2^2$, we obtain the first statement of the theorem by replacing the stepsize $\eta = \eta_0/\sqrt{T}$.

Using the same argument as the proof of Theorem 2, there exists positive constant $\kappa_1, \kappa_2$ such that

$$\mathbb{E}[\|\nabla \mathcal{L}(\theta_t)\|_2] \leq \kappa_1 T^{-1/4} + \kappa_2 \gamma^{K-1} + \|\varepsilon_{K'}\|_2^2 + \frac{L_N\eta_0}{\sqrt{T}}\sigma_{K'}.$$

$$\square$$

