# OpenReview forum: "FSNet: Feasibility-Seeking Neural Network for Constrained Optimization with Guarantees"
_NeurIPS.cc/2025/Conference — NeurIPS 2025 poster_

### Official Review · Reviewer_wKYo · 2025-07-01

**Clarity:** 3
**Significance:** 3
**Originality:** 3
**Rating:** 4
**Confidence:** 3

**Summary:**

The paper introduces FSNet, a novel framework that integrates feasibility-seeking procedures into neural network pipelines to solve constrained optimization problems efficiently and with theoretical guarantees. Unlike traditional neural solvers or NN-based surrogate models that often produce infeasible solutions, FSNet ensures constraint satisfaction by solving a differentiable, unconstrained optimization subproblem that minimizes constraint violations. This feasibility-seeking step is integrated into both the training and inference stages. The authors provide theoretical convergence guarantees under smoothness and PL conditions and demonstrate that FSNet achieves near-zero constraint violations with competitive or superior objective values, while being significantly faster than traditional solvers across convex, nonconvex, and nonsmooth problem settings.

**Questions:**

1. The introduction lacks clarity on the specific problem the paper addresses. It would help to present equations (1) and (2) earlier. The high-level idea in lines 29–36 is difficult to follow without knowing the underlying problem and assumptions.

2. The feasibility and convergence guarantees are derived under the PL condition (line 43 and Assumption 1). However, how can the loss landscape induced by a neural network satisfy the PL condition? This assumption seems strong, particularly since PL does not generally hold even for some strongly convex functions.

3. In line 111, it is stated that x represents parameters. It is unclear whether the optimization problems mentioned earlier (e.g., in energy systems, robotics) conform to the structure in equation (1). Clarifying this connection would be helpful.

4. The assumption that x \~ D is drawn from a finite distribution is strong. What happens if D is continuous or infinite? How would this affect the theoretical results and practical performance?

5. Theorem 3 mentions convergence of a sequence to limit points. However, does the convergence of the underlying optimization procedure itself hold? Under what conditions?

6. How does the scalability of FSNet depend on the number of variables, constraints, and the size of distribution D (x~D)? It would be valuable to analyze this both theoretically and through experiments on larger-scale problems.

**Ethical Concerns:**

["NO or VERY MINOR ethics concerns only"]

**Final Justification:**

The rebuttal resolved my questions. However, I do not see the applicability of these questions. It seems tedious to use a neural network to approximate solving optimization questions. I raise my score to 4 as I acknowledge that this paper is well-written in a way that is theoretically rigorous and experimentally sound. I lower my confidence to 3 as I have some concern on the motivation.

**Limitations:**

Yes

**Quality:**

3

**Strengths And Weaknesses:**

The paper proposes FSNet, a neural network framework that integrates a feasibility-seeking step for solving constrained optimization problems. The key innovation lies in formulating an unconstrained, differentiable subproblem that enforces feasibility, enabling end-to-end training while guaranteeing convergence under certain assumptions. The method performs well empirically across a variety of convex, nonconvex, and nonsmooth problems, achieving near-zero constraint violation and competitive or superior objective values, often with large runtime improvements over traditional solvers.

However, the presentation of the problem setup is unclear early in the paper. Equations (1) and (2), which are central to the formulation, are introduced only later, making it difficult to understand the context of the framework during the initial discussion. Furthermore, the theoretical analysis relies on strong assumptions such as the PL condition and boundedness of derivatives. These conditions are difficult to justify in the context of neural networks and general nonconvex optimization, especially given that even strong convexity does not necessarily imply PL. The connection between the general applications discussed in the introduction and the formulation in (1) also lacks clarity. Additionally, the assumption that the input distribution is finite (x ~ D) may be restrictive in practice, and it is unclear how the method would perform or be affected in infinite or continuous distributions. Finally, the scalability of FSNet with respect to problem size and dataset cardinality is not analyzed theoretically or validated through large-scale experiments.

---

> ### Author Rebuttal · Authors · 2025-07-30
>
> Thank you to the reviewer for their comments and ideas for ways to further clarify our paper's content.
> >(W) The presentation of the problem setup is unclear early in the paper. Equations (1) and (2), which are central to the formulation, are introduced only later, making it difficult to understand the context of the framework during the initial discussion.
> (Q) The introduction lacks clarity on the specific problem the paper addresses. It would help to present equations (1) and (2) earlier. The high-level idea in lines 29–36 is difficult to follow without knowing the underlying problem and assumptions.
>
> We will plan to move some of these descriptions earlier in the paper to provide additional clarification and grounding to the discussion.
>
> > (W) The theoretical analysis relies on strong assumptions such as the PL condition and boundedness of derivatives. These conditions are difficult to justify in the context of neural networks and general nonconvex optimization, especially given that even strong convexity does not necessarily imply PL.
> (Q) The feasibility and convergence guarantees are derived under the PL condition (line 43 and Assumption 1). However, how can the loss landscape induced by a neural network satisfy the PL condition? This assumption seems strong, particularly since PL does not generally hold even for some strongly convex functions.
>
> We would like to clarify that the main contribution of this work is to provide **a strong empirical framework for deep learning with constraint enforcement in amortized optimization**. This framework is *backed by* theoretical analysis in order to provide further insight into performance and design choices, but importantly, *does not depend on* the assumptions used within the theoretical analysis. For example, we do not use the L-smoothness and the PL condition as a starting point in designing FSNet. Moreover, our theoretical analysis definitely does not cover all settings where FSNet can work well.  For instance, our experiments evaluate FSNet in settings where the L-smoothness and the PL condition do not hold – notably, the PL condition is generally violated in nonconvex problems, and the L-smoothness condition is violated in both convex and nonconvex SOCP problems and all nonsmooth nonconvex problems– and FSNet performs robustly across all tested scenarios.
>
> In other words, the theoretical analysis is an **additional contribution** compared to prior works providing only empirical methods, rather than being something that our framework is directly dependent on.
>
> As an additional clarification, our theoretical analysis does not assume the PL condition for the loss function or loss landscape of the neural network, but rather for the violation function defined as $ \phi (s;x) :=||h(s;x)||_2^2 + || g^+(s;x) ||_2^2 $, which quantifies the constraint violation. Additionally, if a function is strongly convex, then it satisfies the PL inequality, but not vice versa. Thus, the PL condition is a weaker assumption than the strong convexity. For further reference, please refer to [1].
>
> [1] Karimi, Hamed, Julie Nutini, and Mark Schmidt. "Linear convergence of gradient and proximal-gradient methods under the Polyak-Łojasiewicz condition." Joint European conference on machine learning and knowledge discovery in databases. Cham: Springer International Publishing, 2016.
>
> > (W) The connection between the general applications discussed in the introduction and the formulation in (1) also lacks clarity.
> (Q) In line 111, it is stated that $x$ represents parameters. It is unclear whether the optimization problems mentioned earlier (e.g., in energy systems, robotics) conform to the structure in equation (1). Clarifying this connection would be helpful.
>
> Thank you for highlighting the need to clarify this connection. Below are some illustrative examples, and we will incorporate this discussion into the revised manuscript.
>
> In AC optimal power flow (ACOPF) for power systems, $x$ represents the load demands, and $y$ denotes the optimal power generation at the generators. In building temperature control, $x$ corresponds to the current temperature state of the building, and $y$ represents the control signal for actuators such as heaters and air conditioners. In robot motion planning, $x$ includes the initial and terminal states as well as the sequence of feasible corridors, while $y$ corresponds to the optimal traversal time.
>
> Beyond these, the formulation in equation (1) also captures a wide range of problems in meta-learning, reinforcement learning, optimization, and control. For additional examples, we refer the reader to the tutorial on amortized optimization [1].
>
> [1] Amos, Brandon. Tutorial on amortized optimization. Foundations and Trends® in Machine Learning (2023).
>
> > (W) The assumption that the input distribution is finite ($x \sim \mathcal{D}$) may be restrictive in practice, and it is unclear how the method would perform or be affected in infinite or continuous distributions.
> (Q) The assumption that $x \sim \mathcal{D}$ is drawn from a finite distribution is strong. What happens if D is continuous or infinite? How would this affect the theoretical results and practical performance?
>
> We *do not* assume the distribution $\mathcal{D}$ is finite, nor do we make any assumptions on whether it is continuous or discrete.
> Instead, as standard in machine learning, we assume that we have access to a finite dataset drawn from $\mathcal{D}$, and that the exact form of $\mathcal{D}$ is unknown to the learner. This is a standard assumption in machine learning [1]. If the reviewer can point to any places in the manuscript from which this misunderstanding may have arisen, we will plan to make revisions to the manuscript accordingly to aim to mitigate any confusion.
>
> [1] Shalev-Shwartz, Shai, and Shai Ben-David. Understanding machine learning: From theory to algorithms. Cambridge university press, 2014.
>
> > (W) The scalability of FSNet with respect to problem size and dataset cardinality is not analyzed theoretically or validated through large-scale experiments.
> (Q) How does the scalability of FSNet depend on the number of variables, constraints, and the size of distribution $\mathcal{D}$ ($x \sim \mathcal{D}$)? It would be valuable to analyze this both theoretically and through experiments on larger-scale problems.
>
> Philosophically, FSNet reduces the online solution complexity of the original constrained optimization problem to that of a (much easier) *un*constrained optimization problem – notably, finding a feasible point via the feasibility-seeking step. As a result, the scalability of FSNet is directly related to the scalability of unconstrained optimization, with **all complexity and convergence analyses existing for unconstrained optimization applying directly**. Therefore, we believe that handling problems with thousands or even tens of thousands of decision variables is practically feasible, provided that appropriate scalable solvers are used for the feasibility-seeking step. In fact, FSNet achieves greater speedup relative to traditional solvers as the number of variables and constraints increases, as shown in our results in AC optimal power flow and larger-scale convex problems. For the detailed results, please refer to our response to the last comment of `Reviewer q8xn`.
>
> Regarding the impact of dataset size on the scalability of FSNet, we note that this primarily affects training time, not inference time. As the dataset grows, the time per epoch increases proportionally, leading to a longer total training duration. However, this is generally not a concern, as training is typically performed offline and only once, while the trained model can be used many times during inference. In particular, the overall motivation for the area of amortized optimization is that we can replace some of the expense of online inference with offline costs incurred during training, with the idea that online inference time (rather than offline training time) is the primary bottleneck.
>
> > (Q) Theorem 3 mentions the convergence of a sequence to limit points. However, does the convergence of the underlying optimization procedure itself hold? Under what conditions?
>
> We would like to clarify that Theorem 3 does not mean the convergence of the entire sequence $\{(y_\tau, \hat y_\tau)\}$. Rather, it shows that *if* the sequence admits a limit point $(y^\star, \hat{y}^\star)$, then this limit point is a global minimizer of the original optimization problem (1) under the assumptions stated in the theorem. The only condition required for the existence of a limit point is that the sequence ${(y_\tau, \hat y_\tau) }$ is bounded. In this case, the Bolzano-Weierstrass theorem guarantees the existence of a convergent subsequence, which defines the limit point. Our result is consistent with similar guarantees in classical penalty methods (see [1], p. 471).
>
> To prove convergence of the entire sequence, we would require stronger assumptions, such as strong/strict convexity, which do not hold in our general nonconvex setting. For this reason, we do not extend the theorem to that case.
>
> [1] Bertsekas, Dimitri. Nonlinear Programming. Vol. 4. Athena Scientific, 2016.

---

### Official Review · Reviewer_q8xn · 2025-07-02

**Clarity:** 3
**Significance:** 2
**Originality:** 3
**Rating:** 4
**Confidence:** 3

**Summary:**

In this paper, the Feasibility-Seeking-Integrated Neural Network (FSNet) is proposed to solve optimization problems with complex constraints. Core idea of this method is to introduce a differentiable feasibility search step based on neural network predictions, ensuring that the output solutions satisfy the constraints while enabling end-to-end training. This design not only provides theoretical guarantees of convergence (with exponential convergence under certain conditions), but also demonstrates excellent empirical performance across various types of problems (convex/non-convex, smooth/non-smooth).

**Questions:**

Please refer to the **Weaknesses** part.

**Ethical Concerns:**

["NO or VERY MINOR ethics concerns only"]

**Final Justification:**

The authors have addressed my main concerns.

**Limitations:**

Please refer to the **Weaknesses** part.

**Quality:**

2

**Strengths And Weaknesses:**

**Strengths**
1. Theoretical Rigor and Clarity: A detailed convergence analysis is provided, including exponential convergence for feasibility search and guarantees for end-to-end training, which enhances the credibility of the approach.

2. Sufficient Empirical Evidence: Extensive experiments have been conducted on a variety of complex problems, comparing the performance with traditional solvers as well as other learning methods, and the results are convincing.

3. Good Presentation: The paper is well-written, main body of the paper and the level of details in the appendix are appropriate.

**Weaknesses**
1. Assumption Dependency for Non-Convex Problems: Current convergence guarantees hold under certain assumptions (e.g., L-smoothness, PL condition). However, in practical applications of non-convex problems, these conditions might difficult to ensure. It is suggested to clearly define the scope of applicability of these conditions in the discussion (or limitation) and to look forward to future relaxation schemes.

2. Algorithm Complexity and Practicality: Although the method performs excellently in several problems, the computational cost of the feasibility search step in high-dimensional, large-scale problems still needs attention. I suggest supplementing an analysis of its complexity and the discussion on its application potential in ultra-large-scale problems.

3. Experimental Design and Comparison: The experiments primarily focused on smaller-scale problems (e.g., 100 decision variables). In the future, the scalability of the method could be validated in higher-dimensional, more complex real-world application scenarios.

Overall, I consider this is an interesting paper. If the authors could address the concerns during the rebuttal process, I would be happy to increase my score.

---

> ### Author Rebuttal · Authors · 2025-07-30
>
> Thank you to the reviewer for their careful review and thoughtful comments.
> > Assumption Dependency for Non-Convex Problems: Current convergence guarantees hold under certain assumptions (e.g., L-smoothness, PL condition). However, in practical applications of non-convex problems, these conditions might be difficult to ensure. It is suggested to clearly define the scope of applicability of these conditions in the discussion (or limitation) and to look forward to future relaxation schemes.
>
> We would like to clarify that the main contribution of this work is to **provide a strong empirical framework for deep learning with constraint enforcement in amortized optimization**. This framework is *backed by* theoretical analysis in order to provide further insight into performance and design choices, but importantly, *does not depend on* the assumptions used within the theoretical analysis. For instance, we do not use the L-smoothness and the PL condition as a starting point in designing FSNet. Moreover, our theoretical analysis definitely does not cover all settings where FSNet can work well. In other words, the theoretical analysis is an **additional contribution** compared to prior works providing only empirical methods, rather than being something upon which our framework is directly designed.
>
> In our experiments, we have already evaluated FSNet in settings where the L-smoothness and the PL condition *do not* hold. For example, while the PL condition may be (locally) satisfied in some convex problems, it is generally violated in nonconvex problems, which can have saddle points with vanishing gradients and nonzero function values. Similarly, the L-smoothness condition is violated in both convex and nonconvex SOCP problems due to the presence of $\ell_2$ norms in the constraints, which make the violation function $\phi$ nondifferentiable and thus not L-smooth. In addition, we add the $\ell_2$ norm regularization in the objective function in nonsmooth nonconvex cases (please see Table B.3), which also makes the objective functions not L-smooth. Despite these violations, FSNet performs robustly across all tested scenarios.
>
> Future work includes further strengthening the theoretical analysis to capture more general settings, to better reflect the strong empirical performance across more general settings that we observe in practice. We will plan to add discussion on these points to the paper.
> > Algorithm Complexity and Practicality: Although the method performs excellently in several problems, the computational cost of the feasibility search step in high-dimensional, large-scale problems still needs attention. I suggest supplementing an analysis of its complexity and the discussion on its application potential in ultra-large-scale problems.
>
> We agree that scalability is an important consideration for FSNet, especially in the context of high-dimensional and large-scale problems. To further evaluate this aspect, we have now conducted additional experiments on larger problems; detailed results are provided in our response to the third comment. Notably, in experiments on the AC optimal power flow (ACOPF) problem in power grids, we observe that FSNet's speedup gains actually *increase* with problem size – e.g., a **3**$\times$ speedup on the IEEE 30-bus system (60 variables, 60 equality constraints, 248 inequality constraints) and an **11**$\times$ speedup on the IEEE 118-bus system (236 variables, 236 equality constraints, 1196 inequality constraints). A similar pattern is observed when comparing FSNet's performance on 500-variable convex problems to the 100-variable cases. This trend highlights FSNet’s scalability and is attributable to the fact that FSNet solves an unconstrained optimization problem initialized by the neural network's output, which is generally more efficient and scalable than solving the original constrained optimization problem.
>
> Regarding computational complexity, the primary computational burden of FSNet lies in the feasibility-seeking step. However, since this step involves unconstrained optimization, standard complexity results and scalability techniques from classical unconstrained optimization can be directly applied. For ultra-large-scale problems, it will be important to incorporate techniques commonly used at this scale into FSNet, such as sparse matrix operations, scalable numerical linear algebra, and exploitation of problem structure. Based on our observation that FSNet achieves greater speedup gains in higher-dimensional and larger-scale problems, we expect it to dramatically outperform traditional solvers in ultra-large-scale settings.
>
> In summary, we believe that FSNet can be readily used for large-scale applications given an appropriate solver design and implementation for the feasibility-seeking step. Thank you for pointing out the need for discussion on this point; we will plan to add this to the paper.
> > Experimental Design and Comparison: The experiments primarily focused on smaller-scale problems (e.g., 100 decision variables). In the future, the scalability of the method could be validated in higher-dimensional, more complex real-world application scenarios.
>
> We have now run experiments on FSNet for the AC optimal power flow (ACOPF) problem in power systems, with results under a random seed presented in Table 1 below. FSNet satisfies equality and inequality constraints (numerical violations on a similar order to that of the IPOPT solver). Moreover, the optimality gap remains below 1$\%$ in both test cases. In terms of runtime in the sequential setting, FSNet significantly outperforms IPOPT, achieving **3**$\times$ speedups on the IEEE 30-bus system and **11**$\times$ on the IEEE 118-bus system. The penalty method, on the other hand, despite its fast inference, suffers from high constraint violations, particularly in terms of maximum violations on the IEEE 118-bus case. Overall, this result is consistent with our previous findings on convex and nonconvex problems in the original manuscript.
>
> *Table 1: Test results of FSNet on 2000 instances of ACOPF problems.*
> | Method|Eq. Vio.|Ineq. Vio.|Opt. Gap (%)|Runtime (s)|
> |-|-|-|-|-|
> ||Mean (Max)|Mean (Max)|Mean / Min (Max)|Sequential|
> |||**IEEE 30-bus**:| $n=60, n_\text{eq}=60, n_\text{ineq}=248$||
> | IPOPT Solver| 6.7e-7 (5.8e-6)| 5.4e-9 (1.1e-6)|--/--|51.53|
> | **FSNet** |1.1e-5 (5.1e-5)|1.3e-8 (2.9e-6)| 0.86 / 0.05 (3.63)|16.80|
> | Penalty|1.5e-3 (6.2e-3)|7.1e-6 (1.7e-3)|-5.91 / -9.90 (-1.69)|0.74|
> |||**IEEE 118-bus**:| $n=236, n_\text{eq}=236, n_\text{ineq}=1196$||
> |IPOPT Solver|1.2e-6 (2.8e-5)|3.1e-8 (1.1e-5)|--/--|392.83|
> |**FSNet**|2.1e-5 (4.9e-4)|9.9e-7 (4.1e-4)|0.78 / -2.66 (3.78)| 34.01|
> |Penalty|4.4e-3 (0.144)|2.9e-5 (0.259)|0.57 / -3.41 (1.61)|0.85|
>
> For larger-scale problems, we have now evaluated FSNet on convex problems with 500 variables, 200 equality constraints, and 200 inequality constraints. This scale matches or exceeds those considered in recent studies (e.g., [1-4]). The results are presented in Table 2 below. In the batch setting, FSNet achieved speedups of **188**$\times$ (QP) and **147**$\times$ (QCQP), which are notably higher than the speedups obtained in smaller problems with 100 variables—namely, 31$\times$ and 104$\times$, respectively (see Table B.1). In the sequential setting, FSNet achieved speedups of **8**$\times$ (QP) and **143**$\times$ (QCQP), compared to 0.8$\times$ and 6$\times$ in the corresponding 100-variable cases. For SOCP, running the 500-variable experiment requires significantly more time. We have already run it for over a day, but were unable to complete it before the rebuttal deadline. We estimate that solving 2,000 instances in the sequential setting will take approximately 3.5 days, and we will update the results as soon as they become available. Based on our estimation, we believe that FSNet will exhibit similarly increasing speedup gains in the SOCP case, as observed in the QP and QCQP cases. This trend is consistent with our findings on ACOPF problems and supports the observation that FSNet becomes increasingly advantageous in larger-scale settings.
>
> *Table 2: Test results of FSNet on 2000 instances of smooth convex problems with 500 variables.*
> | Method|Eq. Vio.|Ineq. Vio.|Opt. Gap (%)|Runtime (s)|
> |-|-|-|-|-|
> ||Mean (Max)|Mean (Max)|Mean / Min (Max)|Batch (Sequential)|
> ||||**Convex QP**: $n=500, n_\text{eq}=200, n_\text{ineq}=200$||
> |Solver|1.9e-9 ± 2.9e-10  (1.2e-6 ± 9.8e-8)|1.4e-6 ± 9.8e-8 (9.3e-4 ± 1.3e-4)|--/--|112.962 ± 0.378 (1394.024 ± 94.868)|
> |**FSNet**|1.0e-4 ± 2.5e-6 (4.4e-3 ± 8.2e-4)|4.5e-6 ± 1.5e-7 (4.2e-4 ± 9.0e-5)|0.066 ± 2.7e-4 / 0.041 ± 6.6e-4 (0.119 ± 4.9e-3)|0.599 ± 1.8e-3  (163.558 ± 1.027)|
> ||||**Convex QCQP**: $n=500, n_\text{eq}=200, n_\text{ineq}=200$||
> |Solver|1.3e-5 ± 4.3e-7 (3.1e-4 ± 7.7e-5)|5.0e-2 ± 9.9e-4 (2.1e-1 ± 7.3e-3)|--/--|5508.419 ± 44.220 (33362.147 ± 107.952)|
> |**FSNet**|2.7e-4 ± 1.4e-6 (3.7e-4 ± 3.3e-7)|9.9e-7 ± 4.5e-8 (1.0e-5 ± 4.9e-7)|0.143 ± 1.3e-3 / 0.087 ± 2.5e-3 (0.328 ± 0.034)|37.291 ± 0.237 (232.837 ± 1.229)|
> ||||**Convex SOCP**: $n=500, n_\text{eq}=200, n_\text{ineq}=200$||
> |**FSNet**|2.7e-4 ± 1.4e-6 (3.7e-4 ± 2.1e-6)|0.0 ± 0.0 (0.0 ± 0.0)|0.730 ± 0.027 / 0.469 ± 0.246 (1.124 ± 0.048)|14.313 ± 0.073 (177.890 ± 1.098)|
>
> [1] Priya L. Donti, David Rolnick, and J Zico Kolter. DC3: A learning method for optimization with hard constraints. In International Conference on Learning Representations, 2021.
>
> [2] Enming Liang, Minghua Chen, and Steven H Low. Homeomorphic projection to ensure neural-network solution feasibility for constrained optimization. Journal of Machine Learning Research, 2024.
>
> [3] Park, Seonho, and Pascal Van Hentenryck. Self-supervised primal-dual learning for constrained optimization. Proceedings of the AAAI Conference on Artificial Intelligence, 2023.
>
> [4] Youngjae Min, Anoopkumar Sonar, and Navid Azizan. Hard-constrained neural networks with universal approximation guarantees. arXiv preprint arXiv:2410.10807, 2024.

---

> > ### Author Response · Authors · 2025-08-05
> >
> > Thank you again for your review of our paper. Please let us know if you have any remaining questions or clarifications, which we would be happy and eager to address during the reviewer-author discussion period.

---

> > > ### Comment · Reviewer_q8xn · 2025-08-06
> > >
> > > I thank the authors for their efforts in the rebuttal. I raise the score from 3 to 4.

---

### Official Review · Reviewer_d4Uk · 2025-07-03

**Clarity:** 3
**Significance:** 3
**Originality:** 3
**Rating:** 4
**Confidence:** 3

**Summary:**

This paper introduces FSNet, a Feasibility-Seeking Neural Network framework aimed at solving constrained optimization problems efficiently while providing guarantees on constraint satisfaction and convergence. The approach integrates a differentiable feasibility-seeking step into both the training and inference pipelines of standard neural networks, minimizing constraint violations via a separate unconstrained optimization problem, which is solved iteratively starting from the NN predictions. Theoretical analyses demonstrate guarantees on exponential convergence to feasible points under standard smoothness and PL conditions, and convergence of the NN parameter optimization under SGD.

**Questions:**

Please list up and carefully describe questions and suggestions for the authors, which should focus on key points (ideally around 3–5) that are actionable with clear guidance. Think of the things where a response from the author can change your opinion, clarify a confusion or address a limitation. You are strongly encouraged to state the clear criteria under which your evaluation score could increase or decrease. This can be very important for a productive rebuttal and discussion phase with the authors.

1. Can the authors provide quantitative analysis or further comments on FSNet’s performance when the smoothness or PL conditions are significantly violated—are there clear failure cases, or does the method degrade gracefully?
2. How sensitive are the results (constraint violation and optimality gap) to the penalty weight $\rho$ and unrolled step count $K$ in both training and inference?
3. Are there foreseeable bottlenecks or challenges when scaling FSNet to thousands or tens of thousands of decision variables? Is the feasibility-seeking optimization step a limiting factor?

**Ethical Concerns:**

["NO or VERY MINOR ethics concerns only"]

**Final Justification:**

My score is already positive, and I am happy to maintain it.

**Limitations:**

yes

**Paper Formatting Concerns:**

None.

**Quality:**

3

**Strengths And Weaknesses:**

**Strengths:**

1. FSNet is not limited to a specific problem class; it is applicable to both convex and nonconvex, smooth and nonsmooth constrained optimization problems. The feasibility-seeking module is conceptually general and practically flexible.
2. FSNet achieves large speedups over solvers (up to 1000x in some cases), showing practical value for real-time or large-scale deployments.

**Weaknesses:**

1. The benchmarks are synthetic, following protocols from previous works, but there is a lack of real, application-driven case studies (e.g., energy systems, supply chains, robotics), which would better validate the practical impact and generalizability.
2. The current method does not exploit specific structures of certain problem classes, which leads to FSNet being outperformed by highly optimized solvers. There is limited discussion on how to adapt FSNet to leverage such structure for further efficiency gains.
3. The method’s performance and guarantees depend on tuning parameters such as penalty $\rho$, convergence thresholds, and number of unrolled iterations in the feasibility-seeking step.

---

> ### Author Rebuttal · Authors · 2025-07-30
>
> Thank you to the reviewer for their support and thoughtful comments.
> > The benchmarks are synthetic, following protocols from previous works, but there is a lack of real, application-driven case studies (e.g., energy systems, supply chains, robotics), which would better validate the practical impact and generalizability.
>
> We have now run experiments on FSNet for the AC optimal power flow (ACOPF) problem in power systems. For the detailed results table, please refer to the third comment of `Reviewer q8xn`. FSNet satisfies equality and inequality constraints (numerical violations on a similar order to that of the IPOPT solver). Moreover, the optimality gap remains below 1$\%$ in both test cases. In terms of sequential runtime, FSNet significantly outperforms IPOPT, achieving **3**$\times$ speedups on the IEEE 30-bus system and **11**$\times$ on the IEEE 118-bus system. The penalty method, on the other hand, despite its fast inference, suffers from high constraint violations, particularly in terms of maximum violations on the IEEE 118-bus case. Overall, this result is consistent with our previous findings on convex and nonconvex problems in the original manuscript.
> > The current method does not exploit specific structures of certain problem classes, which leads to FSNet being outperformed by highly optimized solvers. There is limited discussion on how to adapt FSNet to leverage such a structure for further efficiency gains.
>
> Thank you for pointing out the need for discussion on this point. We believe that FSNet's slower sequential performance on the convex QP problem compared to OSQP is partly due to its implementation in PyTorch using Python, without speed optimization. In contrast, OSQP and other solvers are implemented in fast programming languages with a lot of solver-specific performance optimizations. We expect that, if FSNet were implemented with similar performance optimizations and in the same programming languages as these solvers, it could outperform OSQP. This is because solving the unconstrained feasibility-seeking problem used within FSNet is inherently cheaper and more efficient than solving the original constrained optimization problem.
>
> We aim to design the FSNet framework to have broad applicability and thus avoid incorporating problem-specific structures within this initial general framework. However, there are indeed several straightforward directions to integrate such properties into FSNet (and we are in fact working on some of these in follow-on work for specific real-world problems such as AC optimal power flow). For instance:
> - Problem structure can guide the design of the solver used for the feasibility-seeking step. By exploiting the form or sparsity of constraint functions, a faster solver can be tailored to the problem.
> - In some cases, a subset of decision variables determines the full solution. FSNet can leverage this by having the neural network predict only this subset. The feasibility-seeking step can then work only with this subset, and then the full solution can be recovered after that. This reduces the problem dimension and leads to additional speedups. This approach is related to the variable elimination used in DC3.
>
> We will add a discussion on this point to the paper.
> > (W) The method’s performance and guarantees depend on tuning parameters such as penalty $\rho$, convergence thresholds, and number of unrolled iterations in the feasibility-seeking step.
> (Q) How sensitive are the results (constraint violation and optimality gap) to the penalty weight $\rho$ and unrolled step count $K$ in both training and inference?
>
> We agree that the sensitivity to hyperparameters is an important aspect to understand. In our appendix, we provide the experimental results to evaluate the performance of FSNet under different penalty weights and varying numbers of unrolled steps. Overall, we observe that FSNet is not highly sensitive to these choices and maintains consistent performance across settings. We summarize the findings below.
> - **Penalty weight** (Appendix B.5): The total constraint violation and optimality gap remain stable across different values of the penalty weight. However, FSNet tends to run faster with larger penalty weights. This is because higher penalties encourage the neural network to predict points closer to the feasible set, reducing the number of iterations needed in the feasibility-seeking step.
> - **Number of unrolled steps in forward pass**: In our experiments, the L-BFGS solver terminates either when it reaches a maximum of 50 iterations or when the constraint violation function $\phi$ falls below a threshold (typically $1e{-8}$ to $1e{-10}$). The number of unrolled iterations actually varies depending on the difficulty of each problem instance. In our experiments, we observe that the L-BFGS solver often meets the stopping threshold in fewer than 50 iterations. This threshold is a key parameter that directly controls the degree of constraint satisfaction. To investigate the sensitivity of FSNet to the number of unrolled iterations, we instead vary the stopping threshold and show the result in the table below. As expected, using a smaller threshold leads to tighter constraint satisfaction but requires more computation. Conversely, with a larger threshold, the solver may terminate early, and the neural network can produce solutions outside the feasible set that still have low objective values. This behavior can result in a negative optimality gap, as observed with a threshold of $1e{-4}$.
>
> *Table: Test results of FSNet on 2000 instances of convex QP problems with different stopping thresholds for the feasibility-seeking step.*
> |Threshold|Eq. Vio.|Ineq. Vio.|Opt. Gap (%)|Runtime (s)|
> |-|-|-|-|-|
> ||Mean (Max)|Mean (Max)|Mean / Min (Max)|Batch|
> |1e-4|0.045 (0.103)|6.7e-3 (0.017)|7.9e-3 / -0.021 (0.058)|0.071|
> |1e-6|4.2e-3 (0.133)|1.8e-4 (8.4e-4)|0.022 / 7.7e-3 (0.056)|0.098|
> |1e-8|3.8e-4 (1.9e-3)|1.7e-5 (1.4e-4)|0.021 / 7.6e-3 (0.057)|0.126|
> |1e-10|3.4e-5 (1.9e-4)|7.2e-7 (9.1e-6)|0.018 / 6.9e-3 (0.054)|0.174|
>
> - **Number of unrolled steps in backward pass** (Appendix B.4): We also evaluated the performance of FSNet under different numbers of unrolled iterations in the feasibility-seeking steps included in the computational graph for backpropagation (please see Table B.4 in the Appendix). We find that excluding all iterations results in higher optimality gaps, indicating that the neural network fails to learn the optimal solutions. However, differentiating through 10–50 unrolled steps leads to consistent performance, with optimality gaps remaining below 1$\%$.  This makes selecting the number of unrolled steps for the backward pass relatively easy in practice.
>
> > Can the authors provide quantitative analysis or further comments on FSNet’s performance when the smoothness or PL conditions are significantly violated—are there clear failure cases, or does the method degrade gracefully?
>
> In our experiments, we have already evaluated FSNet in settings where the L-smoothness and the PL condition do not hold. For example, while the PL condition may be (locally) satisfied in some convex problems, it is generally violated in nonconvex problems, which can have saddle points with vanishing gradients and nonzero function values. Similarly, the L-smoothness condition is violated in both convex and nonconvex SOCP problems due to the presence of $\ell_2$ norms in the constraints, which make the violation function $\phi$ nondifferentiable and thus not L-smooth. In addition, we add the $\ell_2$ norm regularization in the objective function in nonsmooth nonconvex cases (please see Table B.3), which also makes the objective functions not L-smooth. The ACOPF experiment provided earlier in this response is another example of a setting where L-smoothness and the PL condition are significantly violated. Despite these violations, FSNet performs robustly across all tested scenarios.
>
> We would also like to emphasize that we do not use the L-smoothness and the PL condition as a starting point in designing FSNet. While the main contribution of our paper is the introduction of a strong empirical framework for deep learning with constraint enforcement in amortized optimization, the theoretical analysis is intended to provide *additional insight* into how and why the proposed framework works under certain conditions, and definitely does not cover all settings where FSNet can work well. As such, FSNet is not limited to the specific assumptions used in the theoretical analysis, and our experimental results demonstrate that it performs well even when these conditions are violated.
> >Are there foreseeable bottlenecks or challenges when scaling FSNet to thousands or tens of thousands of decision variables? Is the feasibility-seeking optimization step a limiting factor?
>
> Philosophically, FSNet reduces the online solution complexity of the original constrained optimization problem to that of a (much easier) *un*constrained optimization problem – notably, finding a feasible point via the feasibility-seeking step.
> As a result, the scalability of FSNet is directly related to the scalability of unconstrained optimization, with all toolkits and intuitions available for unconstrained optimization applying directly. Therefore, we believe that handling problems with thousands or even tens of thousands of decision variables is practically feasible, provided that appropriate scalable solvers are used for the feasibility-seeking step.
>
> For this reason, we do not consider the feasibility-seeking step to be a limiting factor. In fact, as shown in our results in ACOPF (in the first comment), FSNet achieves greater speedup relative to traditional solvers as the number of variables and constraints increases. This is because unconstrained optimization is often faster, more scalable, and easier to parallelize than its constrained counterparts. Thus, FSNet can become even more advantageous in large-scale problem settings.

---

> > ### Author Response · Authors · 2025-08-05
> >
> > Thank you again for your review of our paper. Please let us know if you have any remaining questions or clarifications, which we would be happy and eager to address during the reviewer-author discussion period.

---

> > ### Comment · Reviewer_d4Uk · 2025-08-07
> >
> > Thank you to the authors for the detailed and comprehensive rebuttal. Your responses, including the new experiments on ACOPF and the sensitivity analysis, have thoroughly addressed my questions.
> >
> > My score is already positive, and I am happy to maintain it. I appreciate the productive discussion.

---

### Official Review · Reviewer_C6cv · 2025-07-03

**Clarity:** 3
**Significance:** 3
**Originality:** 3
**Rating:** 5
**Confidence:** 4

**Summary:**

This paper proposes FSNet, a neural network framework that integrates a differentiable feasibility-seeking mechanism to solve constrained optimization problems efficiently. A feasibility-seeking procedure is performed via vanilla gradient update. Unlike traditional solvers that are computationally intensive or learning-based methods that often produce infeasible solutions, FSNet explicitly minimizes constraint violations during inference, ensuring solution feasibility. The approach supports end-to-end training and offers theoretical guarantees on convergence and feasibility.

**Questions:**

See weaknesses.

**Ethical Concerns:**

["NO or VERY MINOR ethics concerns only"]

**Final Justification:**

The authors' response has adequately addressed my concerns, and therefore I have increased my score.

**Limitations:**

Yes.

**Paper Formatting Concerns:**

No concerns.

**Quality:**

3

**Strengths And Weaknesses:**

Strengths: The authors present their ideas clearly, with well-written content and detailed mathematical derivations.

Weaknesses: The proposed gradient step-based method is similar to optimizer-based constraint satisfiability layers, such as CvxpyLayers, OptNet, SATNet, GLinSAT, etc. Some of these layers use interior point method to solve the projection problem while some of them use gradient-based first-order methods. Actually, the proposed method FSNet can be regarded as a projection layer whose objective is to minimize the constraint violation and the resulting (convex/non-convex) optimization problem is solved by the gradient-based method. One potential gap between the theorem in this paper and the reality is that, it is usually hard to decide Lipschitz constant in the actual computation. Instead of everytime tuning the stepsize by hand, will it be better to automatically approximate the Lipschitz constant especially in convex cases? Besides, if there is some potential way to tackle the stepsize in non-convex cases, that will be much better.

---

> ### Author Rebuttal · Authors · 2025-07-30
>
> Thank you to the reviewer for their support and thoughtful comments.
>
> > The proposed gradient step-based method is similar to optimizer-based constraint satisfiability layers, such as CvxpyLayers, OptNet, SATNet, GLinSAT, etc. Some of these layers use the interior point method to solve the projection problem while some of them use gradient-based first-order methods. Actually, the proposed method FSNet can be regarded as a projection layer whose objective is to minimize the constraint violation and the resulting (convex/non-convex) optimization problem is solved by the gradient-based method.
>
> We would like to emphasize that FSNet is conceptually different from the mentioned frameworks. The projection-based approach solves a constrained distance-minimization problem to map an infeasible point back onto the feasible set. Unlike the projection-based approaches, **FSNet gets rid of the notion of distance minimization and instead directly minimizes the magnitude of constraint violations**. Our approach offers advantages related to scalable computation and broad applicability as follows:
>
> - The feasibility-seeking step is faster, more scalable, and easier to parallelize than the projection layer, primarily because it involves solving an unconstrained problem, whereas projection requires solving a constrained optimization problem. In our experiments, the projection layer using qpth (i.e., OptNet) is even slower than directly solving the original problem with OSQP. As shown in Table B.1, the projection approach takes 13.71 seconds to solve QP problems in parallel, compared to 3.802 seconds with OSQP and only 0.121 seconds with FSNet. This means FSNet is approximately 100$\times$ faster than using the projection layer. In addition, we also attempted to implement CvxpyLayers for general convex problems, but found it significantly slower than qpth for QPs and too slow to be practical for other convex problems.
>
> - The differentiable optimization layers the reviewer references only support linear or convex constraints, e.g., CvxpyLayers (disciplined convex constraints), qpth (linear constraints), SATNet (semidefinite constraints), and GLinSAT (linear constraints). While nonconvex constraints could in theory be handled by implementing custom differentiable nonconvex projection layers, we opted not to do this given the poor computational performance of projection methods in the convex settings we tested. In contrast, FSNet can readily and scalably handle both convex and nonconvex constraints, enabled by our key idea of directly seeking a feasible point via an easier-to-solve optimization subproblem.
> - The DC3 framework is another alternative to projection in the implicit layers literature that the reviewer references. However, it performed significantly worse than FSNet in our experiments.
>
> Another major point of difference is that previous works on this topic are typically validated and interpreted primarily through experiments, and lack theoretical insights (e.g., [1], [2], [3]).
> In contrast, within FSNet, we provide both an empirical framework with strong computational performance and generality, and also a theoretical analysis of feasibility and convergence. Put otherwise, FSNet goes beyond the previous approaches by offering both empirical effectiveness and theoretical guarantees.
>
> [1] Tanneau, Mathieu, and Pascal Van Hentenryck. Dual Lagrangian learning for conic optimization. Advances in Neural Information Processing Systems, 2024.
>
> [2] Priya L. Donti, David Rolnick, and J Zico Kolter. DC3: A learning method for optimization with hard constraints. In International Conference on Learning Representations, 2021.
>
> [3] Park, Seonho, and Pascal Van Hentenryck. Self-supervised primal-dual learning for constrained optimization. Proceedings of the AAAI Conference on Artificial Intelligence. Vol. 37. No. 4. 2023.
>
> > One potential gap between the theorem in this paper and the reality is that, it is usually hard to decide Lipschitz constant in the actual computation. Instead of everytime tuning the stepsize by hand, will it be better to automatically approximate the Lipschitz constant especially in convex cases? Besides, if there is some potential way to tackle the stepsize in non-convex cases, that will be much better.
>
> While the Lipschitz constant is commonly used within theoretical analyses of first-order methods (e.g., [1]), in practice, when actually running these methods, the Lipschitz constant is not used directly. In our experiments, we follow the literature in using practical methods as follows:
> - For the feasibility-seeking step, we implement L-BFGS with a backtracking line search to determine the stepsize for sufficient decrease at each optimization iteration. This solver is applied across both convex and nonconvex problems without requiring additional tuning.
> - For the NN training, we use the AdamW optimizer and perform a few trials to choose a proper learning rate. The same learning rate is used across all problems. It is also standard practice in machine learning to use grid search to tune learning rates when needed.
>
> We would also like to emphasize that we do not use the assumptions in the theorems (e.g., the L-smoothness and the PL condition) as a starting point in designing FSNet. While the main contribution of our paper is the introduction of a new method for constraint enforcement in deep learning, the theoretical analysis is intended to provide additional insight into performance and design choices, and definitely does not cover all settings where FSNet can work well. As such, FSNet is not limited to the specific assumptions within the theoretical analysis.
>
> In our experiments, we have already evaluated FSNet in settings where the L-smoothness and the PL condition do not hold. For example, while the PL condition may be (locally) satisfied in some convex problems, it is generally violated in nonconvex problems, which can have saddle points with vanishing gradients and nonzero function values. Similarly, the L-smoothness condition is violated in both convex and nonconvex SOCP problems due to the presence of $\ell_2$ norms in the constraints, which make the violation function $\phi$ nondifferentiable and thus not L-smooth. In addition, we add the $\ell_2$ norm regularization in the objective function in nonsmooth nonconvex cases (please see Table B.3), which also makes the objective functions not L-smooth. Despite these violations, FSNet performs robustly across all tested scenarios.
>
> In summary, the Lipschitz constant is used solely in the theoretical analysis; in practice, FSNet operates effectively without requiring it. Moreover, the theoretical analysis in this paper is aimed at offering insight into how and why FSNet works under certain conditions, and is an **additional contribution** compared to prior works providing only empirical methods. Importantly, the assumptions in theorems do not restrict the applicability of FSNet, as our experimental results demonstrate that FSNet performs robustly well even when the theoretical assumptions are violated.
>
> [1] Garrigos, Guillaume, and Robert M. Gower. "Handbook of convergence theorems for (stochastic) gradient methods." arXiv preprint arXiv:2301.11235 (2023).

---

> > ### Author Response · Authors · 2025-08-05
> >
> > Thank you again for your review of our paper. Please let us know if you have any remaining questions or clarifications, which we would be happy and eager to address during the reviewer-author discussion period.

---

> > ### Comment · Reviewer_C6cv · 2025-08-05
> > **Official Comment by Reviewer C6cv**
> >
> > I would like to thank the authors for their response. The authors' response has adequately addressed my concerns, and therefore I have increased my score.

---

### Decision · Program_Chairs · 2025-09-17

**Decision:**

Accept (poster)

**Comment:**

The paper proposes a NN based approach to solving constrained optimization problem. The approach is to have an NN network predict a candidate optimum, and then via sequential GD steps moved towards the constraint set and then an overall loss is applied balancing the optimality of the solution and the closeness of the original prediction and the feasibility seeking solution.

The paper is very well written and presents the ideas in a clean and concise way. The reviewer discussion was highly productive with reviewers appreciating the more practice oriented results and clarifications to the questions raised. I recommend acceptance.